# Evidence from pupillometry, fMRI, and RNN modelling shows that gain neuromodulation mediates task-relevant perceptual switches

Gabriel Wainstein[1†], Christopher J Whyte[1,2†], Kaylena A Ehgoetz Martens[3], Eli J Müller[1,2], Vicente Medel[1,4], Britt Anderson[3], Elisabeth Stöttinger[5], James Danckert[3], Brandon R Munn[1,2†], James M Shine[1,2*†]

[1]Brain and Mind Center, The University of Sydney, Sydney, Australia; [2]Center for Complex Systems, The University of Sydney, Sydney, Australia; [3]The University of Waterloo, Waterloo, Canada; [4]Latin American Brain Health (BrainLat), Universidad Adolfo Ibáñez, Santiago, Chile; [5]Hochschule Fresenius, Köln, Germany

**\*For correspondence:**
mac.shine@sydney.edu.au

[†]These authors contributed equally to this work

**Competing interest:** The authors declare that no competing interests exist.

## eLife Assessment

This **valuable** article explores the idea that transient modulations of neural gain promote switches between distinct perceptual interpretations of ambiguous stimuli. The authors provide **solid** evidence for this idea by pupillometry (an indirect proxy of neuromodulatory activity), fMRI, neural network modelling, and dynamical systems analyses. The highly integrative nature of this approach is rare in the field.

**Abstract** Perceptual updating has been hypothesised to rely on a network reset modulated by bursts of ascending neuromodulatory neurotransmitters, such as noradrenaline, abruptly altering the brain's susceptibility to changing sensory activity. To test this hypothesis at a large-scale, we analysed an ambiguous figures task using pupillometry and functional magnetic resonance imaging (fMRI). Behaviourally, qualitative shifts in the perceptual interpretation of an ambiguous image were associated with peaks in pupil diameter, an indirect readout of phasic bursts in neuromodulatory tone. We further hypothesised that stimulus ambiguity drives neuromodulatory tone, leading to heightened neural gain, hastening perceptual switches. To explore this hypothesis computationally, we trained a recurrent neural network (RNN) on an analogous perceptual categorisation task, allowing gain to change dynamically with classification uncertainty. As predicted, higher gain accelerated perceptual switching by transiently destabilising the network's dynamical regime in periods of maximal uncertainty. We leveraged a low-dimensional readout of the RNN dynamics to develop two novel macroscale predictions: perceptual switches should occur with peaks in low-dimensional brain state velocity and with a flattened egocentric energy landscape. Using fMRI, we confirmed these predictions, highlighting the role of the neuromodulatory system in the large-scale network reconfigurations mediating adaptive perceptual updates.

## Introduction

The overwhelming majority of neurons in our brains have only indirect interactions with the external world. This means that the identity of sensory inputs is inherently ambiguous (*Bogacz, 2017*; *Flounders et al., 2019*; *Friston, 2005*; *Hohwy, 2013*; *Clark, 2013*). The equivocal nature of perceptual

input is overcome by incorporating prior information about the causal structure of the world into sensory inferences. This is clearly evidenced in laboratory experiments that present participants with sensory inputs that offer two equally valid yet mutually exclusive perceptual interpretations (e.g. the Necker cube illusion and binocular rivalry): in these ambiguous scenarios, observers periodically switch between mutually exclusive percepts (*Hohwy et al., 2008*; *Alais and Blake, 2005*; *van Ee, 2005*).

Outside of conditions of extreme perceptual ambiguity, perceptual awareness is remarkably stable, suggesting that the nervous system can rapidly (and flexibly) identify the best 'match' between visual data and a stable (likely known) stimulus category (*Hohwy et al., 2008*; *Hohwy, 2012*). Importantly, this process of combining ambiguous sensory input with prior information must be dynamic: adaptive behaviour requires that the relative reliability of prior information and current sensory input are made suitably contextually dependent (*Moran et al., 2013*; *Shine et al., 2021*; *Parr and Friston, 2018*; *Parr and Friston, 2017*). In ecological settings, the problem is even more pronounced: not only does the reliability of the sensory input vary, the urgency of perceptual decision-making also changes between context (*Thura et al., 2020*; *Bogacz et al., 2006*; *Murphy et al., 2016*).

Neuroimaging studies investigating perceptual updating and switches have typically identified a distributed set of regions within the cerebral cortex (*Stöttinger et al., 2015*; *Weilnhammer et al., 2017*). These cortical regions are presumed to play a role in attentional shifts driving switches in perceptual contents by selectively boosting activity within the relevant circuits (*Weilnhammer et al., 2017*; *Reynolds and Heeger, 2009*; *Desimone and Duncan, 1995*). This interpretation is complemented by behavioural evidence showing that attention plays a prominent role in determining the contents of perception in bistable perception tasks where competition is not resolved at low-levels of the visual hierarchy (*Meng and Tong, 2004*; *Dieter and Tadin, 2011*). Similarly, computational models of perceptual decision-making typically consist of winner-take-all competition between cortical populations (*Wong and Wang, 2006*; *Eckhoff et al., 2011*; *Wang, 2002*). Yet, the ability to flexibly respond to ambiguous visual inputs according to changing task demands is a feature that is present across phylogeny (*Cisek, 2019*) and hence is present in a wide variety of animals that have poorly developed cerebral cortices (*Carter et al., 2020*). Indeed, phasic change in the highly conserved ascending arousal system have been linked to moment-by-moment adaptive updates in the relative weighting of prior information, sensory input, and the urgency of the perceptual decision process through neuromodulatory-mediated alterations in neural gain (*Sales et al., 2019*; *Vincent et al., 2019*; *Jordan and Keller, 2023*; *Murphy et al., 2014*).

The ascending neuromodulatory system, and specifically the noradrenergic locus coeruleus (LC), is well-suited to modulate the large-scale, brain state switches required to flexibly alter perceptual contents (*Szabadi, 2018*; *Briand et al., 2007*). While the cell body of the LC is located in the brainstem, the nucleus sends projections throughout the central nervous system, wherein its axons release noradrenaline, which in turn modulate the excitability of targeted regions (*Shine et al., 2021*). In previous work, it has been argued that the phasic release of noradrenaline from the LC acts as a 'network reset' signal, which effectively disrupts ongoing processing, and hence allows animals to reconfigure their ongoing neural dynamics towards more salient (and hopefully, behaviourally relevant) processes (*Sara, 2009*; *Jacob and Nienborg, 2018*; *Bouret and Sara, 2005*). This mechanism is of critical importance in ecological contexts in which an animal needs to be able to both focus on the current task in an exploitative mode (such as foraging), while being able to rapidly modify its internal, attentional, and behavioural state when required (e.g. if resources are depleted, or in the presence of a predator).

Preliminary evidence in the context of bistable perception has shown that when a stimulus is task-relevant, pupil diameter (a non-specific and indirect readout of phasic LC activity [*Joshi and Gold, 2020*; *Samuels and Szabadi, 2008*; *Pfeffer et al., 2022*; *de Gee et al., 2020*] and neuromodulatory tone) is tightly linked to switches in the content of perception (*Murphy et al., 2014*; *Einhäuser et al., 2008*; *Reimer et al., 2016*). In line with this, recent modelling has shown that linking perceptual updates to fluctuations in neuromodulatory tone recapitulates the phasic-tonic firing rate pattern known to characterise LC spiking dynamics and improves performance in reinforcement learning tasks (*Sales et al., 2019*). Thus, whilst the LC could plausibly mediate perceptual switches in a task-relevant setting, we still need a more robust test of this hypothesis.

Based on previous work (*Sara, 2009*; *Jacob and Nienborg, 2018*; *Bouret and Sara, 2005*; *Shine, 2019*; *Wainstein et al., 2021*), and the projections of the LC to many of the regions implicated in

whole-brain imaging studies of perceptual uncertainty (*Samuels and Szabadi, 2008*; *de Gee et al., 2014*; *de Gee et al., 2017*), we hypothesised that task-related perceptual switches can be modulated by phasic bursts of LC activity, which act as a 'network reset' (*Bouret and Sara, 2005*), flattening the whole-brain energy landscape (*Munn et al., 2021*) and thus allowing cortical dynamics to evolve into a new state thereby changing the contents of perception.

To test this hypothesis, we leveraged a cognitive task designed to investigate switches in perceptual categorisation (*Stöttinger et al., 2016*). We observed that pupil diameter peaked at the point of the perceptual switch and predicted their timing. We then trained a recurrent neural network (RNN) to perform an analogous change detection task. Based on previous modelling and theory, we allowed the gain of the activation function (an established mechanism for the action of noradrenaline on the cerebral cortex; *Shine et al., 2021*; *Shine et al., 2018*; *Servan-schreiber, 2016*) to vary as a function of the uncertainty in the pretrained network's perceptual categorisation. This revealed that heightened gain facilitated earlier perceptual switches by transiently destabilising the network's dynamics under conditions of maximal uncertainty. Further analyses translated these neural dynamics into two predictions that could be tested in fMRI data (*Stöttinger et al., 2015*; *Stöttinger et al., 2016*): (1) heightened gain increases the velocity of low-dimensional neural trajectories around perceptual switches, and (2) it flattens the energy landscape of the neural state space (*Munn et al., 2021*). Overall, our results support the hypothesis that phasic bursts of neuromodulatory activity act as a 'network reset' (*Sara, 2009*; *Bouret and Sara, 2005*), dynamically disrupting stable network states and facilitating switches in perceptual categorisation. This reset mechanism highlights the role of neuromodulatory systems in transiently reorganising network dynamics to enhance flexibility and adaptability in response to uncertainty.

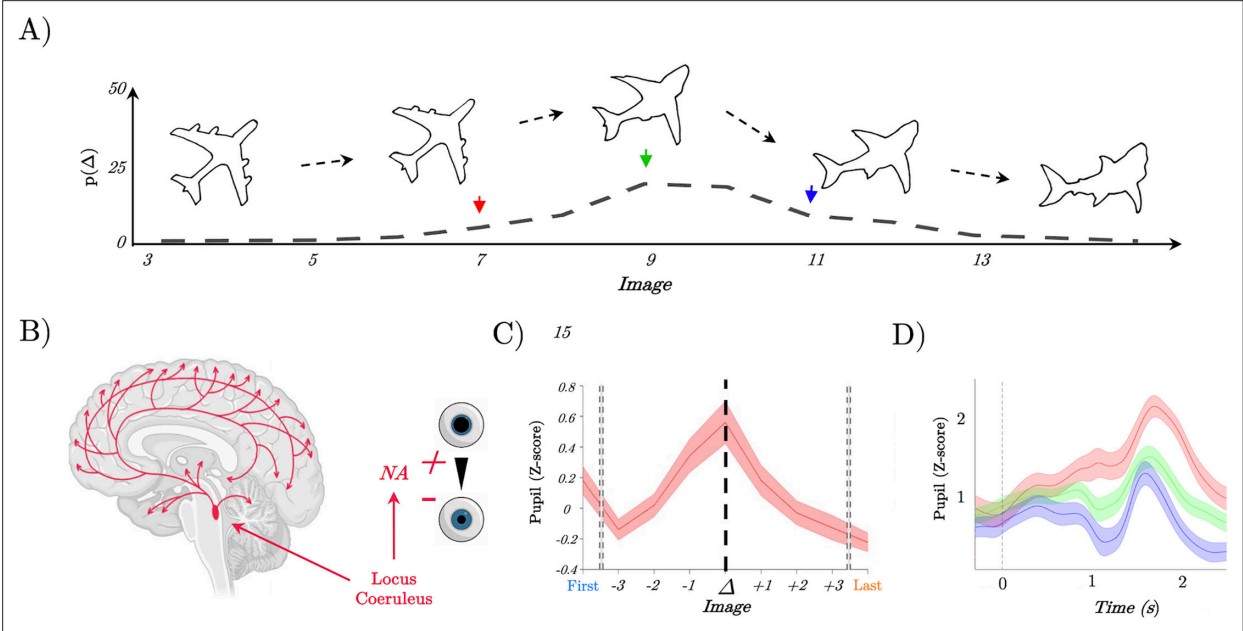

**Figure 1.** Pupil diameter tracks perceptual change. (**A**) Example trial showing the continuous change from a stable image (plane) into a shark; Lower: the probability of detecting a switch (Δ) as a function of Image – most switches occur around the mid-point, but not exclusively so, leading to our prediction of heightened locus coeruleus activity at the switch point. (**B**) Representation of the locus coeruleus (red), its diffuse projections to the whole-brain network and its link to pupil dilation. (**C**) Pupil diameter group average evoked response time locked to the perceptual change (dark line, *t*=0). We observed an increase of the pupillary response that peaked at the perceptual change. (**D**) Group average of evoked pupillary responses to image switches – red represents the faster response when the switch occurs at image 6; green indicates a medium response with the switch at image 8; and blue denotes the slowest response with the switch at image 10.

## Results

### Evoked pupil dilations coincide with the resolution of perceptual ambiguity

To assess the role of the ascending arousal activity during task performance, we analysed a dataset of 35 participants who performed an ambiguous figures task whilst simultaneously recording pupil diameter with an eye tracker device (SR Research, 1000 Hz). Briefly, the task consisted of a set of continuously transforming images that transition from an initial object (e.g. a shark) into a second object (e.g. a plane), while preserving basic psychophysical attributes (*Figure 1A*). Crucially, even though the task stimuli change incrementally and linearly, with maximal ambiguity at the mid-point of each trial (the peak of the dotted line curve in *Figure 1A*), awareness of a change in the stimulus is known to 'pop out', often at different times on each trial (*Stöttinger et al., 2016*). When these perceptual switches occurred, subjects were instructed to change the button they were pressing, thus indicating a change in perceptual interpretation across stimuli. Participants viewed 20 unique sets of images, each of which morphed from a starting image into a second image through 15 equally spaced intermittent stages (*Figure 1A*). For each participant, we identified the first and last time they viewed a sequence of images, as well as the three images leading up to and following an identified perceptual switch, irrespective of the categories associated with each specific object switch. The rest of the analyses in this article are organised around this perceptual transition.

Given the known (admittedly non-specific) relationship between LC activity and the dynamics of pupil diameter (*Joshi et al., 2016*; *Figure 1B*), we were able to test the hypothesis that neuromodulatory tone is associated with perceptual switching. The linear nature of the morphing procedure meant that luminance levels (which could otherwise bias pupil diameter; *Szabadi, 2018*; *Joshi and Gold, 2020*; *Samuels and Szabadi, 2008*) were kept constant across all trials. Additionally, motor preparation was controlled by requiring subjects to press a button on each image (indicating the content of their perception). Mapping all blink-corrected, filtered, and normalised trials over time.

We observed a clear increase in the phasic pupillary response approximately three trials before participants switched to a new perceptual category, potentially reflecting the onset of increased ambiguity towards a new object (*Figure 1C*). This response peaked at the point of the perceptual switch, corresponding to the maximum pupil diameter (*Figure 1C*). Further analysis revealed a significant increase in the mean pupil response starting three images before the change point (mean $\beta$=0.22; $t_{(32)}$ = 8.02, p=2.3 × 10$^{-19}$), before returning to baseline levels.

Next, we sought to elucidate the relationship between ascending arousal, quantified by pupil diameter, and the temporal dynamics of perceptual shifts on a trial-by-trial basis. Given the pivotal role of the LC in modulating sensory processing and perceptual switches (*Figure 1B*), we hypothesised that the speed of a perceptual switch would correlate with neuromodulatory tone. Specifically, we predicted that trials with faster perceptual switches would be associated with an increase in pupil diameter, while slower switches would correspond to a decrease.

To test this prediction, we performed a two-level linear model analysis. The peak pupil diameter observed during the perceptual switch was designated as the independent variable, and the trial on which the perceptual shift was reported served as the regressor for each subject. To control for potential confounds, such as impulsive premature responses, and address reduced statistical power in extreme response epochs (both early and late), we limited our analysis to responses within two images from the median switch point (9±2; 84.1% of total trials). At the group level, we conducted a one-tailed *t*-test on the regressors from the linear model. As expected, we observed an inverse relationship between evoked pupil diameter and the trial marking the perceptual switch (mean $\beta$=–0.19, $t_{(27)}$=–2.6452, p=6.7 × 10$^{-3}$, SD=0.3880). Earlier responses showed a positive relationship with higher evoked pupil diameter during the switch epoch (*Figure 1D*, red), whereas later responses were associated with a more constricted pupil (*Figure 1D*, blue). In summary, these results provide indirect evidence for our hypothesis that ascending neuromodulation – such as LC activity – is associated with the speed of perceptual switches.

## Computational evidence for neuromodulatory-mediated perceptual switches in a recurrent neural network

Our initial results provided confirmatory evidence implicating the neuromodulatory tone of the ascending arousal system in perceptual switches. There is evidence, however, suggesting that simply changing stimulus categories can also induce similar pupillary dilations (*Einhäuser et al., 2008*; *Hupé et al., 2009*, *Kloosterman et al., 2015*). What we need, therefore, is a more mechanistic means of both framing and testing our network reset hypothesis in the context of perceptual switching. Along with others (*Kosciessa et al., 2021*; *Eldar et al., 2013*; *Urai et al., 2017*; *Kloosterman et al., 2015*), we have used a combination of computational modelling (*Shine et al., 2018*; *Müller et al., 2020*), neurobiological theory (*Shine et al., 2021*), and multi-model neuroimaging (*Reynolds and Heeger, 2009*; *Carter et al., 2020*; *Murphy et al., 2014*; *Szabadi, 2018*) to suggest that noradrenaline alters neural gain (*Servan-schreiber, 2016*; *Aston-Jones and Cohen, 2005*), which in turn affects inter-regional communication flattening the energy landscape traversed by the brain's dynamics allowing the brain state to jump between perceptual attractors more easily. Whether these signatures of large-scale network reconfiguration are mechanistically related to network reset remains an important and open question.

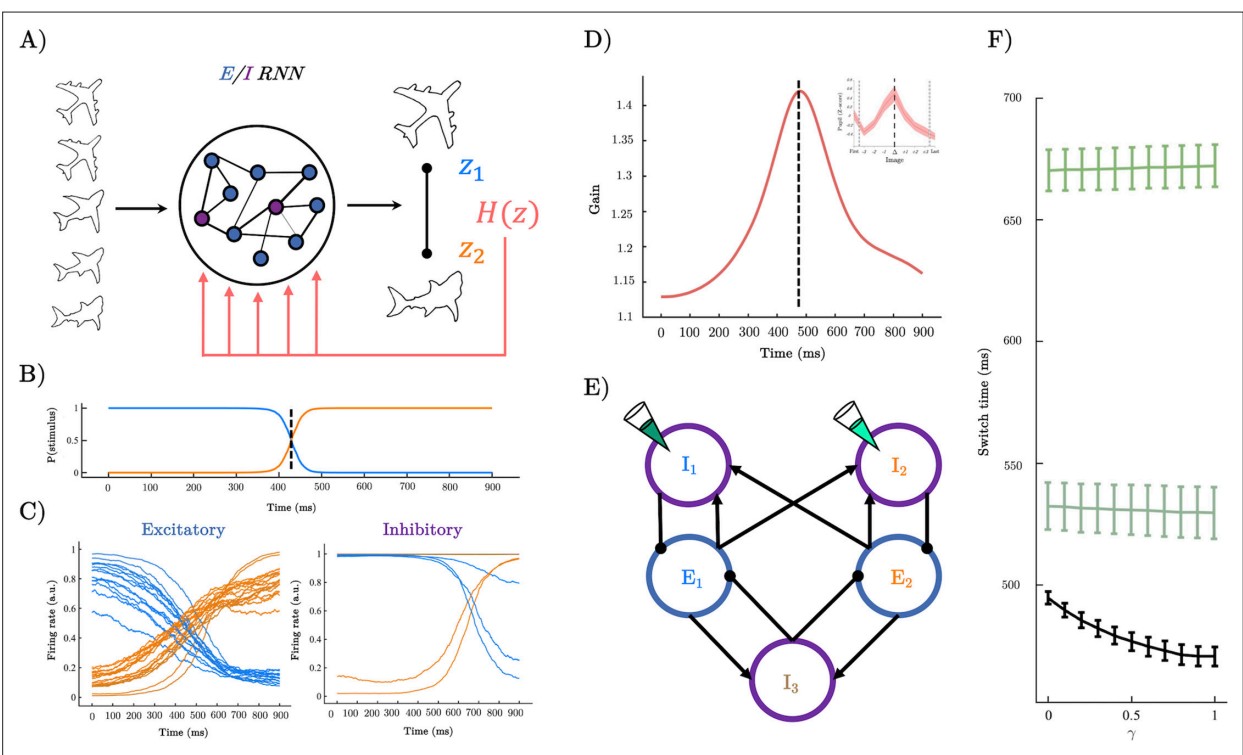

**Figure 2.** A recurrent neural network (RNN) model of perceptual switching. (**A**) We trained a continuous time E/I RNN to categorise linearly changing inputs representing two discrete categories (e.g. output $z_1$ and output $z_2$). (**B**) Softmax of network outputs on example trial with $\gamma = .6$, dotted line shows the timing of the perceptual switch. (**C**) Following training, the firing rate of the excitatory units was clearly separated into two stimulus-selective clusters – those that responded maximally to $u_1$ (blue) and those that respond maximally to $u_2$ (orange). Inhibitory units demonstrated a similar modular clustering but were sorted by the selectivity of the excitatory units they inhibited. (**D**) Dynamics of gain on example trial with $\gamma = .6$ which peaks close to the perceptual switch (inset shows similarity to pupil diameter). (**E**) Simplified network structure implied by selectivity analysis. Excitatory units (blue) form two stimulus-selective modules. Each excitatory cluster is inhibited by a cluster of inhibitory units and a third non-selective inhibitory population. Pipette shows lesion targets. (**F**) Switch time as a function of $\gamma$ magnitude (i.e. magnitude of uncertainty forcing). Lower black line shows a speeding effect of heightened $\gamma$ (and therefore heightened gain at the perceptual switch). Teal lines show switch time for lesions to the inhibitory population targeting the initially dominant population (dark teal upper), and lesions to the inhibitory population selective for the stimulus the input is morphing into (light teal middle).

The online version of this article includes the following figure supplement(s) for figure 2:

**Figure supplement 1.** Switch time as a function of gain for the network simulated with static (i.e. constant) gain.

**Figure supplement 2.** Left selectivity of excitatory units (0: completely selective for $u_1$, 1: completely selective for $u_2$).

To test whether our hypothesised neuromodulatory mechanism could recapitulate the behaviour we observed in the ambiguous figures task, we trained 50 continuous time RNNs constrained to respect Dale's law (i.e. 80/20 split of purely excitatory/inhibitory units; *Song et al., 2016*; *Yang and Wang, 2020*) to perform a perceptual change detection task analogous to the task performed by our participants (*Figure 2A*). The input and readout weights were constrained to be purely excitatory and only the firing rate of excitatory units contributed to the readout (*Song et al., 2016*; see 'Methods').

Each network was provided with a two-dimensional input $u(t) = [u_1, u_2]^T$ with each column representing the 'sensory evidence' for each of the two stimulus categories (*Figure 2A*). The task lasted for 1 s of simulation time (we used a shorter time period for the simulation than the empirical task so that we could keep the integration step relatively small, making the training and simulations more numerically tractable): to mimic the linear transition between image categories in our task, each trial began with maximum evidence for one of the two categoriesand minimum evidence for the other (e.g. $u_1 = 1, u_2 = 0$), and then linearly changed the evidence over the course of each trial such that by the final time-step the evidence for each category had switched (e.g. $u_1 = 0, u_2 = 1$). At each time point, the network was trained to output a categorical response indicating which input dimension had a higher value (*Figure 2B*). Following training, all networks achieved near-perfect behavioural accuracy (0.97±0.02).

We next sought to test our hypothesis about the role of neural gain in perceptual switches. In previous work, we (and others) have argued that the impact of neuromodulators (such as NA) on population-level activity can be approximated by steepening (or flattening) the sigmoid activation function, thus mimicking the effect NA has on neuronal excitability by liberating intracellular calcium stores and/or opening (or closing) voltage-gated ions channels (*Shine et al., 2021*; *Wainstein et al., 2022*). As a first test of this hypothesis, we manipulated the *gain* of the sigmoid activation function for all units in the network across a range of gain values (0.5–1.5) in a static manner. As predicted, increased gain (red; corresponding to heightened adrenergic tone) leads to earlier 'perceptual switches' in the network output whereas low gain caused later switches (*Figure 2—figure supplement 1*).

Having confirmed that static manipulations of gain alter the speed of perceptual switches, we constructed a more precise test of our hypothesis. Specifically, inspired by previous theoretical and experimental work showing that sensory prediction errors (i.e. transient increases in perceptual uncertainty) lead to phasic bursts in the noradrenergic LC (*Sales et al., 2019*; *Jordan and Keller, 2023*), we made gain time dependent with dynamics governed by a linear ODE with a forcing term proportional to the uncertainty (i.e. the entropy $H(z) = \sum_i p(z)_i \ln(p(z)_i)$) of the network's readout (*Figure 2A*).

$$\tau \frac{dg}{dt} = g_{tonic} - g + \gamma H(z)$$

When the network's readout becomes uncertain approaching the perceptual switch (i.e. has high entropy), gain increases in a phasic manner (with magnitude $\gamma$), and in the absence of the forcing, gain decays exponentially to its tonic value ($g_{tonic} = 1$). This modification resulted in gain dynamics reminiscent of the participant's pupil diameter (*Figure 2D*), and crucially, the speed of perceptual switches increased with the magnitude of the uncertainty-driven forcing term ($\gamma$; *Figure 2F*).

Having confirmed our hypothesis that increasing gain as a function of the network uncertainty increased the speed of perceptual switches, we next sought to understand the mechanisms governing this effect starting with the circuit level and working our way up to the population level (c.f. Sheringtonian and Hopfieldian modes of analysis; *Barack and Krakauer, 2021*). Because of the constraint that the input and output weights were strictly positive, we could use their (normalised) value as a measure of stimulus selectivity. Inspection of the firing rates sorted by input weights revealed that the networks had learned to complete the task by segregating both excitatory and inhibitory units into two stimulus-selective clusters (*Figure 2C*). As the inhibitory units could not contribute to the networks read out, we hypothesised that they likely played an indirect role in perceptual switching by inhibiting the population of excitatory neurons selective for the currently dominant stimulus, allowing the competing population to take over and a perceptual switch to occur.

To test this hypothesis, we sorted the inhibitory units by the selectivity of the excitatory units they inhibit (i.e. by the normalised value of the readout weights). Inspecting the histogram of this selectivity metric revealed a bimodal distribution, with peaks at each extreme strongly inhibiting a stimulus-selective excitatory population at the exclusion of the other (*Figure 2—figure supplement*

*2*). Based on the fact that leading up to the perceptual switch point both the input and firing rate of the dominant population are higher than the competing population, we hypothesised that gain likely speeds perceptual switches by actively inhibiting the currently dominant population rather than exciting/disinhibiting the competing population. We predicted, therefore, that lesioning the inhibitory units selective for the stimulus (i.e. with normalised selectivity >0.5) that is initially dominant would dramatically slow perceptual switches, whilst lesioning the inhibitory units selective for the stimulus the input is morphing into would have a comparatively minor slowing effect on switch times since the population is not receiving sufficient input to take over until approximately half-way through the trial irrespective of the inhibition it receives. As selectivity is not entirely one-to-one, we expect both lesions to slow perceptual switches but differ in magnitude. In line with our prediction, lesioning the inhibitory units strongly selective for the initially dominant population greatly slowed perceptual switches (*Figure 2F*, upper), whereas lesioning the population selective for the stimulus the input morphs into removed the speeding effect of gain but had a comparatively small slowing effect on perceptual switches (*Figure 2F*, lower).

Having found a circuit-level explanation for the speeding effect of gain, we next sought to understand the network's behaviour at a population level by interrogating the parameter space (with dimensions defined by network input and gain) traversed by the network. Unlike standard non-linear dynamical systems with stationary or (very) slowly time-varying parameters, input and gain change rapidly over the course of each trial, dynamically shifting the location and existence of the attractors shaping the network dynamics. Each trial is, therefore, characterised by a trajectory through a two-dimensional parameter space with dimensions corresponding to the gain of the activation function and the mismatch between input dimensions ($\Delta input$).

Based on the selectivity of the network firing rates, we hypothesised that the dynamics were shaped by a fixed-point attractor, whose location and existence were determined by gain and $\Delta input$, and changed dynamically over the course of a single trial (*Beer, 2022*; *Beer, 2000*; *Sussillo, 2014*; *Sussillo and Barak, 2013*). Because of the large size of the network, we could not solve for the fixed points or study their stability analytically. Instead, we opted for a numerical approach and characterised the dynamical regime (i.e. the location and existence of approximate fixed-point attractors) across all combinations of (static) gain and $\Delta input$ visited by the network. Specifically, for each combination of elements in the parameter space $\theta \in R^{gain \times \Delta input}$ we ran 100 simulations with initial conditions (firing rates) drawn from a uniform distribution between [0,1], and let the dynamics run for 10 s of simulation time (10 times the length of the task – longer simulation times did not qualitatively change the results) without noise. As we were interested in the existence of fixed-point attractors rather than their precise location, at each time point we computed the difference in firing rate between successive time points ($\Delta r = \sum_i r_i(t) - r_i(t - \Delta t)$) across the network. For each simulation, we computed both the proportion of trials that converged to a value of $\Delta r$ below $10^{-2}$ giving us proxy for the presence of fixed points, and the time to convergence, giving us a measure of the 'strength' of the attractor.

Across gain values when $\Delta input$ had unambiguous values ($u_1 \gg u_2$ or $u_2 \gg u_1$), the network rapidly converged across all initialisations (*Figure 3A and C–H*). When $\Delta input$ became ambiguous, however, the dynamics acquired a decaying (inhibition-driven) oscillation and on many trials did not converge within the time frame of the simulation. As gain increased, the range of $\Delta input$ values characterised by oscillatory dynamics broadened. Crucially, for sufficiently high values of gain, ambiguous $\Delta input$ values transitioned the network into a regime characterised by high-amplitude oscillations (*Figure 3D and G*). Each trial can, therefore, be characterised by a trajectory through this two-dimensional parameter space, with dynamics shaped by the dynamical regimes of each location visited (*Figure 3A and B*).

When uncertainty had a small impact on gain (low $\gamma$), the network had a trajectory through an initial regime characterised by the rapid convergence to a fixed point where the population representing the initial stimulus dominated whilst the other was silent (*Figure 3C*), an uncertain regime characterised by oscillations with all neurons partially activated (*Figure 3D*), and after passing through the oscillatory regime, the network once again entered a (new) fix-point regime where the population representing the initial stimulus was silent whilst the other was dominant (*Figure 3E*).

For high $\gamma$ trials, the network again started and finished in states characterised by rapid convergence to a fixed point representing the dominant input dimension (*Figure 3F–H*). However, it differed in how it transitioned between these states. Uncertain inputs generated high-amplitude oscillations, causing the network to flip-flop between active and silent states (*Figure 3G*). We hypothesised

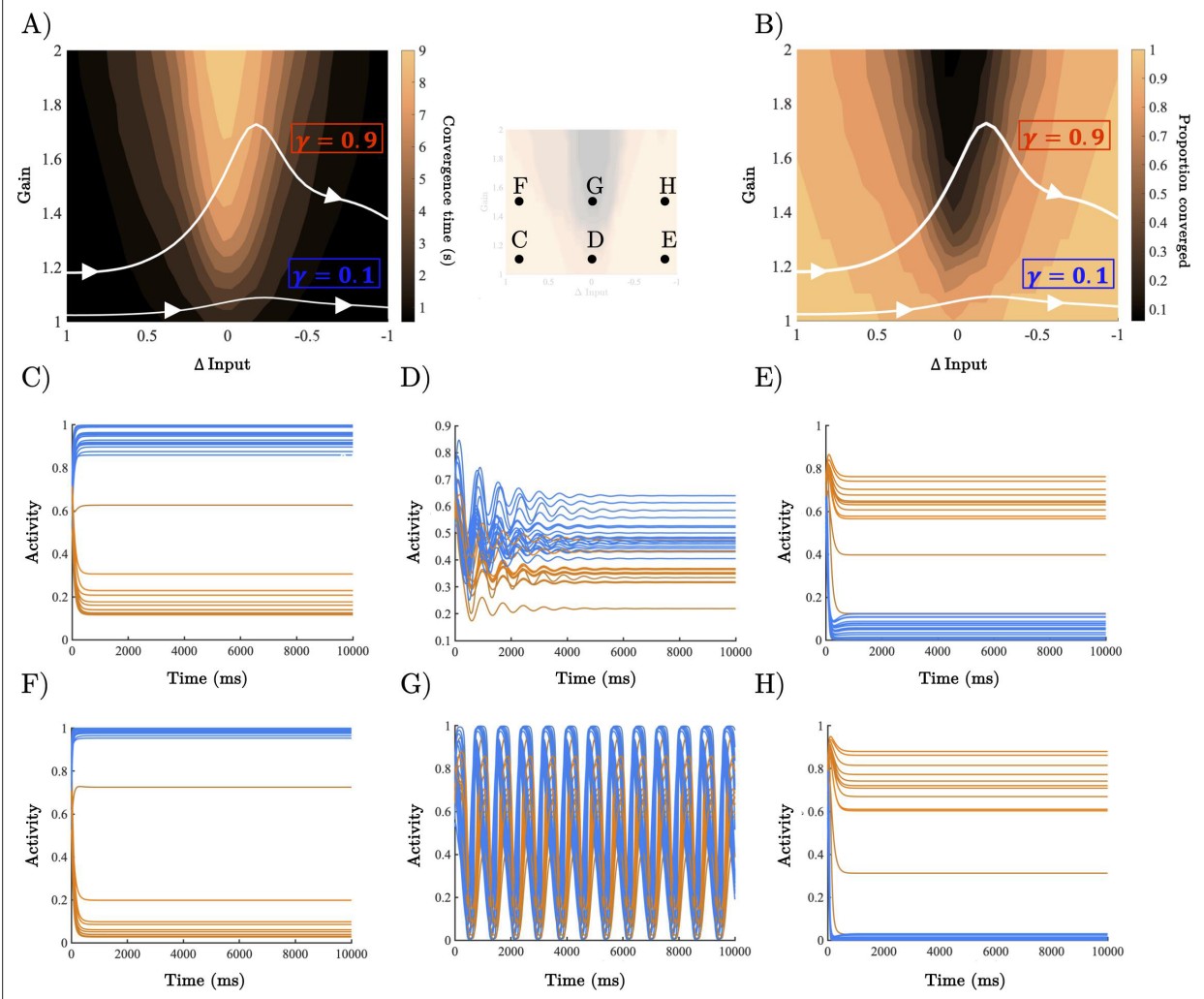

**Figure 3.** Analysis of recurrent neural network (RNN) dynamical regime. (**A**) Contour map of convergence time across the full gain by Δinput parameter space averaged across 100 initialisations with random initial conditions. Example parameter trajectories shown in white for high and low γ trials. (**B**) Contour map of convergence proportion across the full parameter space. (**C–E**) Example dynamics with gain = 1.1 and Δinput ≈ [1, 0],[. 5, .5], and [0,1] r, respectively. (**F–H**) Example dynamics with gain = 1.5 and Δinput ≈ [1, 0],[. 5, .5], and[0,1] , respectively.

The online version of this article includes the following figure supplement(s) for figure 3:

**Figure supplement 1.** Example network dynamics initialised with parameter values well inside the oscillatory regime ($u \approx \left[.5\, .5\right]$, gain = 1.5) with initial conditions determined by the selectivity of each unit.

that, within the task, this mechanism silenced the initially dominant population, while boosting the competing population. To test this, we initialised each network with parameter values well inside the oscillatory regime (u ≈ [. 5 . 5], gain = 1.5) with initial conditions determined by the selectivity of each unit. Excitatory units selective for $u_1$, as well as the associated inhibitory units projecting to this population, were fully activated, whilst the excitatory units selective for $u_2$ (and the associated inhibitory units) were silenced (and vice versa for $u_2 \rightarrow u_1$ trials). As we predicted, when initialised in this state the network dynamics displayed an out-of-phase oscillation where the initially dominant population was rapidly silenced and the competing population was boosted after a brief delay (219 (ms), ±114; *Figure 3—figure supplement 1*).

At the population level, therefore, heightened gain at points of ambiguity accelerates perceptual switches by transiently pushing the dynamics into an unstable regime. This regime replaces the fixed-point attractor representing the input with an oscillatory regime that actively inhibits the currently dominant population and boosts the competing population, before transitioning back to a stable

(approximate) fixed-point attractor representing the new stimulus (*Figure 3F–H* and *Figure 3—figure supplement 1*).

## Large-scale neural predictions of recurrent neural network model

Having confirmed the behavioural component of our gain modulation hypothesis in our model, and characterised both the circuit and population level mechanisms, we next sought to test our hypotheses that the speeding effect of uncertainty-driven gain on perceptual switches is mediated by a flattening of the energy landscape traversed by the network dynamics. Crucially, translating the dynamics of the RNN into an energy-based framework also allowed us to generate a series of predictions that we could later test in functional neuroimaging data.

In recent work (*Munn et al., 2021*, *Taylor et al., 2022*), we have shown that peaks in BOLD within the LC precede large changes in brain state dynamics. Viewed through the lens of dynamical systems theory (*John et al., 2022*) in which the brain is treated as a dynamical system whose state space (i.e. an instantaneous snapshot of the activity of all regions of the system) evolves over time shaped by the presence (or absence) of attractors, the effect of the LC can be conceptualised as akin to lowering the energy barrier required to escape a fixed-point attractor or as a transient injection of kinetic energy via an external force allowing the brain to reach a novel location in state-space (*Munn et al., 2021*). Crucially, there are two complementary viewpoints from which we can construct an energy landscape; the first allocentric (i.e. third-person view) perspective quantifies the energy associated with each position in state space, whereas the second egocentric (i.e. first-person view) perspective quantifies the energy-associated relative changes independent of the direction of movement or the location in state space. The allocentric perspective is straightforwardly comparable to the potential function of a dynamical system but can only be applied to low-dimensional data in settings where a position-like quantity is meaningfully defined. The egocentric perspective is analogous to taking the point of view of a single particle in a physical setting and quantifying the energy associated with movement relative to the particle's initial location. An egocentric framework is thus more applicable, when signal magnitude is relative rather than absolute (see 'Methods and *Figure 4—figure supplement 1* for an intuitive explanation of the allocentric and egocentric energy landscape analysis on a toy dynamical system).

To characterise the energy landscape traversed by the network dynamics, we ran both time-resolved allocentric and egocentric energy landscape analyses. For the allocentric analysis, we first had to reduce the dimensionality of the RNN's dynamics by performing a principal component analysis (PCA) on the concatenated activity of the network at gain = 1. The set of PCs was low-dimensional, with 80.58±6.34% of the variance explained by the first principal component ($PC_1$). Based on this information, we projected the network activity on each trial and for each gain value and timepoint onto the first PC. The resultant low-dimensional trajectories all showed a change in direction around the timepoint of the switch in network output from category 1 to category 2 (and v.v.; *Figure 4A*). This recapitulates a system jumping between attractors, occurring earlier as a function of heightened gain associated with heightened values of $\gamma$ (*Figure 4A*). This switch not only occurred sooner as a function of heightened gain, it also occurred at a higher neural 'speed' with the velocity of the trajectory peaking sharply at the point of the switch under high $\gamma$, whereas the transition between states was comparatively gradual under low $\gamma$ (*Figure 4B*).

With a low-dimensional description of our data in hand, we leveraged the relationship between probability and energy in statistical mechanics to construct a measure of the allocentric energy landscape (*Figure 4C and D*) traversed by the low-dimensional dynamics ($E_{PC1\tau} = ln\left(\frac{1}{P(PC1\tau)}\right)$; see 'Methods' for derivation) with a window size of $\tau = 250\,ms$. Across values of $\gamma$, this revealed a potential-like energy landscape with a minimum that evolved with the currently dominant input dimension. To quantify the effect of gain mediated changes on the allocentric energy landscape, we devised a measure – neural work – of the 'force' exerted on the low-dimensional trajectory by the vector field quantified by allocentric energy landscape at each time point in the trial $W_t = \frac{-dE_t}{dx}s_t$. Where $s_t$ is the displacement of the PC trajectory in each window, and $\frac{dE_t}{dx}$ is the gradient of the energy values computed between the start and end of each window. We found that increasing gain (via increasing $\gamma$) increased the magnitude of work done at turning points of the trajectory analogous to the application of an external force (*Figure 4G*; and equivalent to a change in the dynamical velocity of the landscape, accelerating the change from one perceptual interpretation to another).

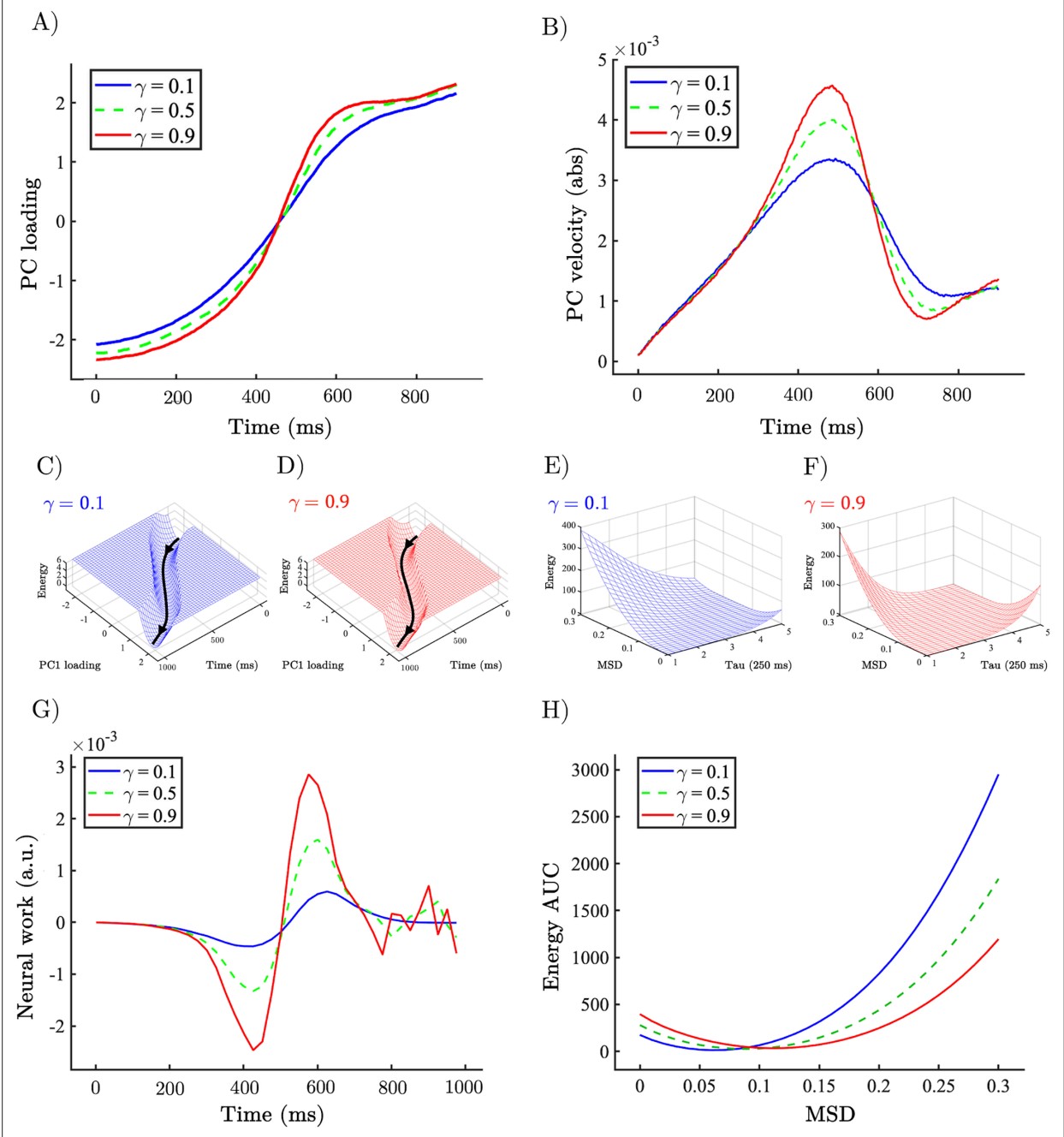

**Figure 4.** Allocentric and egocentric energy landscape dynamics underlying the perceptual speeding effect of heightened gain. (**A**) Example network trajectory projected onto $PC_1$ and averaged across trials for low (0.1; solid blue), medium (0.5; dotted green), and high (0.9; solid red) $\gamma$ for the $u_1 \rightarrow u_2$ condition. (**B**) (abs) Velocity of $PC_1$ trajectories across low (0.1), medium (0.5), and high (0.9) $\gamma$. (**C, D**) Allocentric landscapes for low (0.1; blue) and high (0.9; red) $\gamma$ conditions. Trial-averaged $PC_1$ trajectory shown in black. For purposes of visualisation energy, values > 6 are set to a constant value. (**E, F**) Egocentric landscapes for low (0.1; blue) and high (0.9; red) $\gamma$ conditions. (**G**) (Allocentric) neural work for low (0.1), medium (0.5), and high (0.9) $\gamma$, averaged across networks and conditions. (**H**) Egocentric AUC for low (0.1), medium (0.5), and high (0.9) $\gamma$, averaged across networks and conditions. Note that the time series represent simulations from a model with low noise, and hence did not require error bars.

The online version of this article includes the following figure supplement(s) for figure 4:

**Figure supplement 1.** Energy landscape analyses.

Although explanatory useful in understanding the operation of the RNN, the allocentric landscape is not straightforwardly applicable to non-invasive neuroimaging data. In order to compare our network dynamics to neuroimaging data, and with previous work from our group, we inferred an estimate of the egocentric energy landscape (*Figure 4E and F*) traversed by the dynamics. Specifically, we calculated the mean-squared displacement $MSD_{t,t_0} = \left\langle \left| x_{t_0+\tau} - x_{t_0} \right|^2 \right\rangle_n$ of the firing rate of each unit in the RNN in steps of $\tau = 250\,\mathrm{ms}$, and as we did with the allocentric analysis, calculated the probability – and from this the energy $E_{MSD,\tau} = ln\left(\frac{1}{P(MSD_\tau)}\right)$ – associated with each MSD value and time step. In line with our hypothesis, and with previous work from our group, the energy required for large movements in state space (i.e. large MSD values) decreased as a function of $\gamma$ (*Figure 4E and F*) analogous to the application of an external force transiently increasing the kinetic energy of a particle. To quantify the degree of flattening, we calculated the area under the curve across values of $\gamma$ showing a substantial reduction in the energy associated with large MSD values as a function of heightened $\gamma$ (and therefore gain; *Figure 4H*).

These results reinforce our previous work and clearly demonstrate that the implementation of neuromodulatory-mediated dynamics in the RNN acted in a similar fashion to previously observed patterns in resting-state fMRI (*Cisek, 2019*). In addition, our results confirm that the putative impact of the release of noradrenaline from the LC can change the manner in which brain states evolve over time, facilitating the navigation of otherwise difficult state transitions (*Cisek, 2019*).

## Low-dimensional signature of ambiguity resolution and perceptual change

Having confirmed our hypothesis about the speeding effect of gain in our RNN model, we next sought to test the predictions in the human brain – that is, examining whether the increase in neural speed and the flattening of the energy landscape observed in the RNN were also present in functional neuroimaging data. To this end, we re-analysed an existing BOLD dataset collected while participants performed a similar version of the ambiguous figures task to identify the low-dimensional patterns that occur during the perceptual change.

We were, however, left with a dilemma: RNNs provide a proof-in-principle of how computations can be instantiated in neural networks; however, there are key differences between artificial neural networks and the human brain that require careful consideration (*Richards et al., 2019*; *Doerig et al., 2023*). While both RNNs and the brain are thought to compute through dynamics (*Vyas et al., 2020*), the human brain is comprised of highly specialised neural circuits that have been shaped over evolutionary time to perform a range of highly idiosyncratic functions that matter for adaptive behaviour (*Vyas et al., 2020*), but are not necessarily related to task-switching. So where in the brain should we look for the same low-dimensional signatures we observed in the RNN as a function of gain? Rather than select a particular region a priori, we instead opted for a data-driven approach – PCA – which summarises regional time series concatenated across all subjects and trials into a set of low-dimensional patterns that can then be interrogated in a similar fashion to the activity of the RNN (see 'Methods' for details). Consistent with previous work (*Shine et al., 2019*), a small number of PCs mapped onto distributed regions across the brain (*Figure 5A*) and explained a substantial proportion of the variance observed in the task (PC$_{1-3}$ explained 32% of the total variance).

To isolate the low-dimensional component that best reflected the task (*Figure 1A*), we performed a principal component regression (*Jolliffe, 1982*) that modelled the switch point of each trial using the loadings of the top 3 PCs calculated from fMRI data. PC$_1$ was not selectively aligned with switches, both PC$_2$ and PC$_3$ showed a pronounced, isolated peak around the switch point across trials (*Figure 5B*), with PC$_2$ showing the most robust task-related engagement (*Figure 5B* and *Figure 5—figure supplement 1*). To ensure that these results could not be explained by the spatial autocorrelation inherent within the PC maps, we created a null distribution of regression coefficients calculated using the same statistical model but with block-resampling applied to the switch times in the design matrix. The dotted grey line in *Figure 5B* denotes the 95th percentile of the null distribution, and clearly shows that the engagement of both PC$_2$ and PC$_3$ during the switch point was greater than to be expected by chance. Furthermore, to validate that the perceptual switch was predominantly represented by PC$_2$ and PC$_3$ (*Figure 5D*), we conducted a regression with these two PCs as predictors and the evoked activity derived from the original BOLD time series as the dependent variable. The resulting variance accounted for was 88% ($R^2$=0.88, $\beta$=0.99, p=9.2 × 10$^{-178}$).

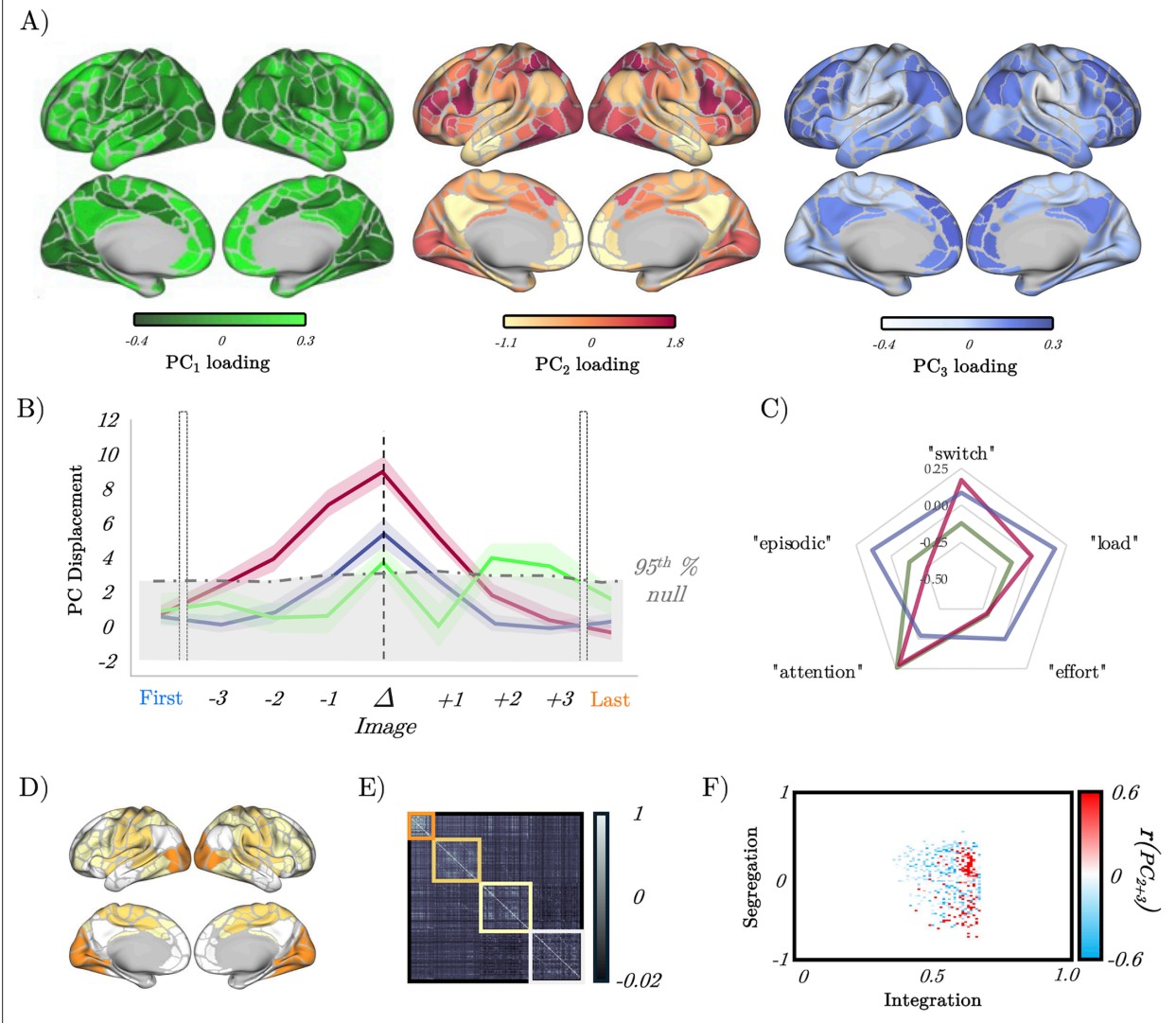

**Figure 5.** Low-dimensional switch-related dynamics and connectivity. (**A**) Spatial loadings of $PC_1$ (green), $PC_2$ (red), and $PC_3$ (blue). (**B**) Mean absolute β loading (solid lines) and group standard error (shaded) of $PC_1$ (green), $PC_2$ (red), and $PC_3$ (blue), organised around the image switch point (Δ) – the dotted grey lines show the 95th percentile of the null distribution of a block-resampling permutation. (**C**) Radar plot showing the partial correlations of $PC_1$ (green), $PC_2$ (red), and $PC_3$ (blue). (**D**) Evoked brain activity of $PC_2 + PC_3$ during the perceptual switch. (**E**) Group averaged functional connectivity and module assignments using a Louvain analysis – three clusters were observed. (**F**) Pearson's correlation between the sum of $PC_2$ and $PC_3$ (per subject) and a joint-histogram comparing Integration (participation coefficient) and segregation (module-degree Z-score); $p<0.05$ following permutation testing.

The online version of this article includes the following figure supplement(s) for figure 5:

**Figure supplement 1.** Energy landscape analyses.

To determine whether $PC_2$ or $PC_3$ was a better index of perceptual switching, we then correlated the spatial loadings of $PC_2$ and $PC_3$ with the spatial map associated with the term 'switching' from a meta-analysis performed on the *neurosynth* database (*Yarkoni et al., 2011*). We observed a significant positive correlation between the map for 'switching' and both $PC_2$ ($r=0.453$, $p=3.041 \times 10^{-18}$) and $PC_3$ ($r=0.115$, $p=0.037$); however, the correlation for $PC_3$ was much lower than $PC_3$, suggesting that $PC_2$ was a better match for 'switching'. The spatial map of $PC_2$ was also positively correlated with other terms putatively associated with the ambiguous figures task (notably, 'effort', 'load', and 'attention'; all $r>0.2$; and not with 'episodic', which was included as a negative control), a partial correlation analysis revealed that $PC_2$ was selectively associated with 'switching' and 'attention' (*Figure 5C*). Given the multifaceted nature of the ambiguous figures task, the convergence between brain maps

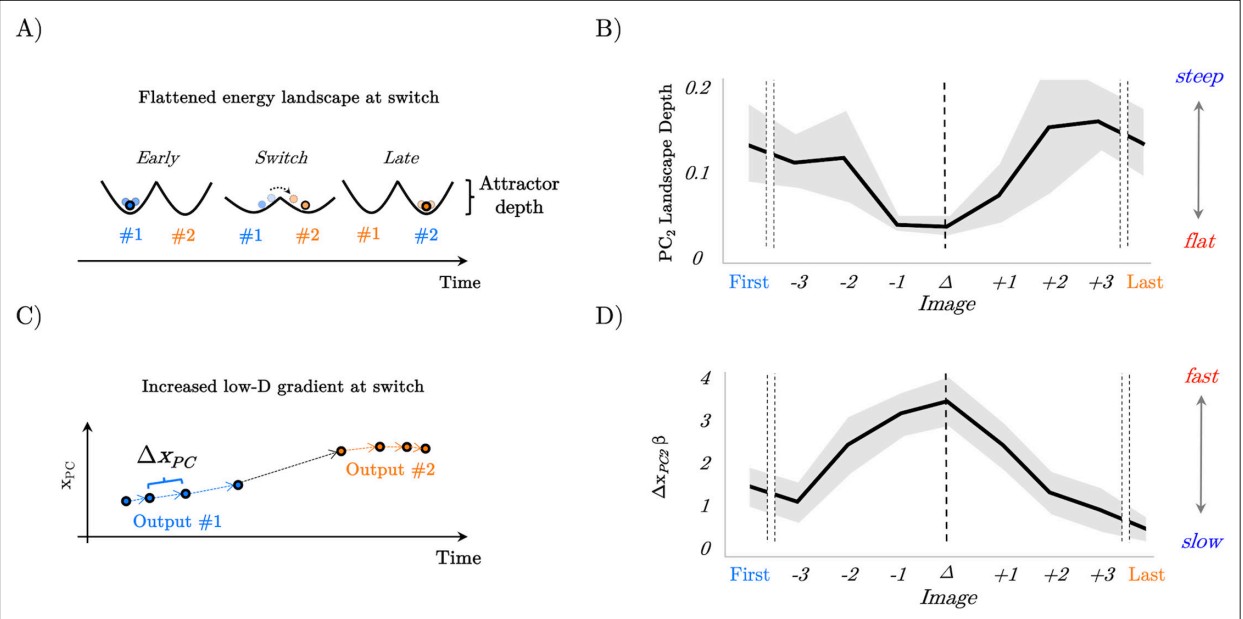

**Figure 6.** Confirmation of model predictions in whole-brain BOLD data. (**A**) analysis of the recurrent neural network (RNN) also predicted that the energy landscape dictating the likelihood of state transitions should be flat (i.e. have a small attractor depth) at the switch point. (**B**) The energy landscape was demonstratively flatter (quantified as surprisal over brain activity displacement) at the switch point. (**C**) By interrogating the low-dimensional trajectories in the RNN, we predicted that there should be a peak in the gradient of the loadings in principal component space at the switch point between output #1 and output #2. (**D**) The gradient ($\Delta x_{PC}$) of the β loading of $PC_2$ as a function of the switch point.

for 'switching', 'attention', and 'effort' was to be expected, and we therefore did not try to dissociate them in further analysis.

Before turning to the predictions of the RNN, we first sought to validate the face validity of focusing on a limited number of principal components. In previous work, we have linked the impacts of NA on systems-level neural dynamics to alterations in network topology (*Munn et al., 2021*; *Shine et al., 2018*; *Shine et al., 2016*), with NA increasing large-scale network integration. Given that PCA naturally captures patterns of covariance between regions, we expected to see that the observed time signatures of PC engagement at the switch point should coincide with similar measures of network integration. To test this hypothesis, we clustered the time-averaged functional connectivity matrix using a hierarchical modular decomposition approach (see 'Methods') – doing so revealed three main clusters (*Figure 5E*). For each participant, we used this matrix and the three clusters to estimate the amount of integration (using the participation coefficient) and segregation (using the module-degree Z-score, see 'Methods') of each region. We then correlated a joint histogram of these measures with the sum of subject-specific regression coefficients for $PC_2$ and $PC_3$ and observed a robust correlation with integration (*Figure 5F*; $p<0.05$ following permutation testing). These results clearly demonstrate the highly convergent nature of PCA and our previous network-based approaches.

## Confirmation of model predictions in whole-brain BOLD data

Based on the patterns observed in the RNN (i.e. those in *Figures 2–4*), we hypothesised that the energy landscape topography would decrease, and the velocity of the low-dimensional brain patterns would peak at the switch point. Given the prominent role in switching, we focused our analysis on the $PC_2$ time series. To estimate the (egocentric) energy landscape, we first estimated the mean displacement of $PC_2$ by averaging the β value around the switch point and then divided this term by the logarithm of the inverse probability of the loading of $PC_2$, which was also inferred from the GLM. Using this approach, we observed that $PC_2$ was maximally displaced at the perceptual change, suggesting that the brain state showed a substantial shift from baseline during the perceptual change. Energy ($\log[1/p_{switch}]$; see 'Methods') showed a U-shaped pattern around the perceptual change point – that is, with a minimum value in the perceptual change along with the first and last images (*Figure 6B–D*). To relate this measure to the energy landscape framework, and to control by the specific displacement

occurring at each image, we then calculated the ratio between energy and the mean displacement (i.e. energy landscape 'depth'; *Figure 6A*). As predicted, the brain state reduced the amount of energy per displacement towards its minimum around the perceptual change (*Figure 6B*). We interpret this set of results as the system flattening the energy landscape, reducing the energy (i.e. higher system changes become more common) required for large displacement values, effectively generating a 'network reset' (*Sara, 2009*; *Bouret and Sara, 2005*; *Sara, 2015*) of the brain state, which ultimately facilitated an updating of the content of perception.

To analyse speed-evoked changes in brain trajectories, we used a GLM to analyse each PC time series as a function of each perceptual switch. Our design matrix included the first and last images seen in each set, along with the three images leading up to the switch, the switch trial itself and the three images following the switch (see 'Methods' for details). This approach thus allowed us to track the low-dimensional signature of the brain through the processing and resolution of perceptual ambiguity. As predicted (*Figure 6C*), we found evidence that $PC_2$ showed a peak in velocity at the change point (*Figure 6D*) providing confirmatory evidence that the low-dimensional brain state dynamics observed in whole-brain fMRI were highly similar to those observed in the trained RNN.

## Discussion

Here, we studied the relationship between the ascending arousal system, low-dimensional neuronal trajectories and energy landscape dynamics during a perceptual switch task. Our results provide evidence that the ascending arousal system is involved in the modulation of dynamic brain state topography during task-relevant perceptual switches. We found that pupil diameter tracked with ambiguity of task stimuli and was directly related to the speed of perceptual switches (*Figure 1*). Next, we confirmed that this process could be replicated in an RNN Model (*Figure 2*) of perceptual change detection where the gain of the activation function was updated dynamically by the uncertainty of the network's classification output (*Figures 2 and 3*). We then used this model to generate two key predictions: around the time of the perceptual switch brain state velocity should peak, and the egocentric energy landscape should flatten which we confirmed in neuroimaging data (*Figures 4–6*). Together, these results suggest that the ascending arousal system facilitates state changes in the content of perception by transiently increasing neural gain – acting in a manner analogous to an external forcing function transiently increasing kinetic energy in the system – flattening the ego-centric energy landscape and thereby reducing the energy needed to reset the system topography in an adaptive and task-dependent manner.

The relationship between perception and pupil diameter found here is consistent with the role of the ascending neuromodulation in cognition and attention (*Aston-Jones and Cohen, 2005*; *Wainstein et al., 2022*). For instance, the LC dynamically changes its activity according to external and cognitive demands imposed on the system (*Joshi et al., 2016*; *Aston-Jones and Cohen, 2005*; *Wainstein et al., 2022*; *Liu et al., 2017*; *Nieuwenhuis et al., 2005*). Importantly, our results extend these findings by suggesting a more precise role for LC-mediated alterations in neural gain. Specifically based on the pupil dynamics in our task and previous experimental and theoretical work, we hypothesised that neural gain should change dynamically as a function of uncertainty (operationalised here as perceptual ambiguity) via the recruitment of the LC (along with other structures in the ascending arousal system), which then subsequently increases brain-wide communication by increasing the gain in targeted brain regions (*Shine et al., 2021*; *Murphy et al., 2016*; *Shine et al., 2018*; *Nieuwenhuis et al., 2005*). In the pupillometry data, pupil diameter (which is an indirect marker of the noradrenergic system; *Szabadi, 2018*; *Joshi and Gold, 2020*) increased as a function of perceptual ambiguity, which rose sharply in the few images prior to the reported perceptual change (*Figure 1D*). Based upon this finding, we then implemented an analogous mechanism in our pretrained RNN by making gain depend upon the entropy of the network's classification which acted as a forcing function transiently increasing gain when the input became ambiguous, which, in line with our hypothesis, lead to earlier perceptual switches. We chose to use an RN instead of a simpler (more transparent) model as we wanted to use the RNN as a means of both hypothesis generation and hypothesis testing. Specifically, unlike more standard neuronal models which are handcrafted to reproduce a specific effect, when building an RNN the modeller only specifies the network inputs, labels, and the parameter constraints (e.g. Dale's law) in advance. The dynamics of the RNN are entirely determined by optimisation. Post-training manipulations of the RNN are not built in, or in any way guaranteed to work, making them

more analogous to experimental manipulations of an approximately task-optimal brain-like system. Confirmatory results are arguably, therefore, a first step towards an in vitro experimental test.

Thus, we provide early empirical and computational evidence that ascending neuromodulatory activity facilitates state changes in perception under conditions of perceptual ambiguity (*Murphy et al., 2014*; *de Gee et al., 2014*; *de Gee et al., 2017*) when a stimulus is task relevant. Importantly, we do not expect that our results will generalise to experimental setting when a stimulus is not task relevant. We can make sense of this computationally by imagining the gain dynamics in our model if we added in a second task-irrelevant condition where at the beginning of each trial the model was given a cue indicating whether it would have to simply 'maintain fixation' or readout the category of the input. In the presence of the task-irrelevant cue, the model would read out the 'maintain fixation' action with high certainty and thus not ramp up gain. We hypothesise therefore that the pupil dynamics observed in the task will depend on participants' task set. Indeed, there is evidence from a recent multistable perception experiment showing that arousal-related changes in pupil dilation disappear when the stimulus is not task-relevant. The authors of the study attribute the arousal-dependent pupil dilation to task execution. This explanation, however, could not explain the ramping of pupil diameter in our task where the participants perform an action on every trial. Instead, based upon the workings of our computational model, we hypothesise that arousal-based changes in pupil diameter are driven by task-set-related uncertainty and thus will depend on task relevance rather than task execution per se.

A core neuroanatomical property of the LC noradrenergic system is that a relatively small number of neurons (~50,000 in an adult human) projects to almost all brain regions (*Samuels and Szabadi, 2008*; *Totah et al., 2019*). This organisation implies that the LC acts as a low-dimensional modulator of the much more high-dimensional cerebral cortex. Subtle changes in the activity of LC can have significant effects on how different brain regions communicate (*Shine et al., 2018*; *Wainstein et al., 2022*; *Liu et al., 2017*; *Zerbi et al., 2019*; *Hansen et al., 2022*; *Hansen et al., 2021*). The mechanism of gain modulation in our model was, likewise, dependent on a low-dimensional process, with the network output altering the gain uniformly across the full network. At a neuronal level, NA increases excitability by liberating intracellular calcium and opening (or closing) voltage-gated ions channels (*Shine et al., 2021*; *Wainstein et al., 2022*). In our model, this global increase in excitability increased the speed of perceptual switches by recruiting inhibitory units to more rapidly actively inhibit the population encoding the initially dominant stimulus. At a population level, the interaction between excitatory and inhibitory units led to the emergence of a gain-dependent oscillatory regime which suppresses the currently active population encoding the initially dominant stimulus and boosts the competing quiescent population. At the scale of the full network, the gain-mediated changes resemble the transient application of an external forcing function pushing the network trajectory in the direction of the new percept which, from the perspective of the allocentric landscape, manifests as a spike in neural work at turning points in the network's low-dimensional trajectory, leading up to and following the perceptual switch. From the egocentric perspective, this is characterised by a flattening of the landscape analogous to an externally driven increase in kinetic energy making large changes in the location of a particle more likely.

In line with the predictions of the RNN in our analysis of the BOLD data, we showed that the velocity of the low- dimensional brain state trajectory most associated with perceptual switching increased significantly during the point of reported perceptual change in comparison (*Figure 5B*), which we interpret as the brain moving from one attractor to another (*Figure 6A*). Importantly, we showed that around the perceptual switch, the energy needed for each unit of change in brain state (i.e. displacement) is smaller than at other points in the task (*Figure 6A and B*). Under the (egocentric) energy landscape framework (*Munn et al., 2021*; *Taylor et al., 2022*), this tells us that the landscape is flattened, and the energy required to transition between states is reduced. Together with the pupillary findings (*Figure 1*), the computation model (*Figures 2–4*), and replication from former results (*Munn et al., 2021*; *Taylor et al., 2022*; *Figure 5E and F*), we propose that the ascending neuromodulatory system is responsible for the large-scale flattening of the egocentric energy landscape which facilitating changes in task-relevant perceptual content.

This work is not without limitations. First, the pupil diameter dataset and the fMRI analysis came from different participants, such that the link between the pupil diameter and the fMRI results is inherently indirect. Moreover, differences in task timing, structure, and instructions between the fMRI and pupil experiments add complexity to interpreting the results. For instance, the fMRI task includes

jittered inter-trial intervals (ITIs) and catch trials, features absent in the pupil task, which presents a more rapid stimulus sequence. These differences may have influenced perceptual switch points and task behaviour across experiments. Additionally, the specificity of the pupil diameter as a marker of the LC activity is under active debate (*Joshi and Gold, 2020*). For instance, there is evidence suggesting a role of the superior colliculus, the dorsal raphe nucleus, and central cholinergic system in driving pupil dilations (*Shine, 2019*; *Cazettes et al., 2021*; *Vyas et al., 2020*; *Shine et al., 2019*). Although there is uncertainty regarding whether these other nuclei are directly related to pupil dilation or only indirectly via their connections with other neural regions and nuclei. Despite this, we believe that our pupillometry dataset captures an important function of the noradrenergic system in cases of task-relevant perceptual ambiguity as there is strong evidence showing that pupil diameter is a reliable marker of noradrenergic activity during evoked cognitive tasks (*Wainstein et al., 2021*; *Zerbi et al., 2019*; *Alnæs et al., 2014*; *Janitzky et al., 2015*; *Reimer et al., 2014*). Additionally, the sample size of our fMRI study makes it difficult to generalise our results. In spite of this, the converging evidence from the pupillometry dataset, the fMRI dataset, and the computational model supports the role of the ascending neuromodulation in mediating task-relevant perceptual switches. Future work is needed both in humans, with higher sample sizes utilising fMRI and eye-tracking recordings, as well as animal studies, to directly modulate and record the LC activity in a task manipulating perceptual uncertainty.

## Conclusion

In summary, we provide computational and empirical evidence for the association between neuromodulation, pupil dilation, and (egocentric) energy landscape flattening in task-relevant perceptual switches. Our results strengthen our understanding of the neurobiological processes underpinning moment-by-moment adaptive changes to perception. Specifically, we suggest that the widespread excitatory projections of the noradrenergic arousal system mediate the systems-level reconfigurations of cortical network architecture (*Totah et al., 2019*; *Zerbi et al., 2019*) via uncertainty-driven alterations in neural gain. This suggests that more highly conserved features of the nervous system may play a role in driving task-relevant switches in the contents of perception.

## Methods
### Overview of empirical data

There were two independent groups analysed in this study: 35 subjects performed a perceptual decision-making perceptual task while pupil diameter was recorded; and a separate group of 17 subjects performed a version of the task adapted for the MRI scanner. Data are available at https://github.com/ShineLabUSYD/AmbiguousFigures (copy archived at *Whyte et al., 2025*).

### Perceptual task

Twenty picture sets were used in which line drawings of common objects morphed over 15 iterations into a different object (*Figure 1A*). Picture sets were selected from a larger set validated in an earlier study (*Stöttinger et al., 2016*). In the original study, participants reported verbally what they saw by typing in the name of the object. This reporting method guaranteed that participants could freely indicate what they saw without being restricted by categories (e.g. forced choice). Picture sets for the current study were selected with the criterion that all sets were perceived categorically in the normative study (i.e. that the majority of participants in the normative study categorised each picture they saw as either the first object or second object in the set; *Stöttinger et al., 2015*). Selecting only the categorically perceived image sets guaranteed that pictures in the middle of the morphing sequence were not simply 'noisier' than pictures at the beginning or end. In other words, the ambiguous images were still easily categorised by participants as either object 1 or object 2. All images were a standard size (316 × 316 pixels) and were displayed on a white background. In addition, in the fMRI study, participants were presented with two kinds of control picture sets to ensure that they were responding to changes in the pictures in the set rather than simply to the position in the set (e.g. always switching after the eighth picture). In these control picture sets, a salient deviating picture was presented either after 3 pictures or after 13 pictures, resulting in an early or late abrupt shift. Those sets served as controls and were not analysed further.

The picture morphing task consisted of five experimental runs. We randomised the order in which the picture sets were presented in each run and kept this randomised order consistent across participants. Picture morphing in each picture set occurred over 15 discrete steps, each corresponding with the acquisition of a whole-brain image. In the fMRI experiment, each picture within a set was presented for 2 s. Pictures were randomly intermixed with eight inter-stimulus intervals (2, 4, 6, or 8 s) during which participants saw a fixation cross. In the eyetracking experiments, each picture was presented for 750 ms, followed by a fixation cross of 2 s. Participants provided their responses in the scanner using two buttons on a four-button Cedrus fibre optic system. In a two-alternative forced-choice task, participants were asked to press the first button when they 'saw the first object' and the second button when they 'saw the second object' – this ensured that there was not a motor confound present on only the switch trials. All participants were ignorant as to the identity of the second object in each picture set. At the end of each set of 15 images, the word END was presented for 2 s to indicate that the next picture set would begin shortly. Participants provided their responses in the fMRI scanner using a Cedrus fibre-optic response system with four buttons. For the two-alternative forced-choice task, participants were instructed to press the first button when they 'saw the first object' and the second button when they 'saw the second object'. This design ensured that motor responses were not confounded with perceptual switches as responses occurred on both switch and non-switch trials. Importantly, participants were not informed about the identity of the second object in each picture set beforehand. At the end of each sequence of 15 images, the word 'END' was displayed for 2 s to signal the conclusion of that picture set and the imminent start of the next one.

## Participants

A total of 17 (six males) neurologically healthy participants with normal or corrected to normal vision took part in the fMRI study (mean age 27.65±8.01). Fifteen were right-hand dominant. A separate cohort of 35 participants performed the task while simultaneous pupil diameter was recorded using an eye tracker device (SR Research, 1000 Hz). None of the participants had a history of brain injury. Participants received $30 for their participation. All participants provided informed consent prior to participation. The research protocol was approved by the Office of Research Ethics at the University of Waterloo and the Tri-Hospital Research Ethics Board of the Region of Waterloo in Ontario, Canada.

## Pupillometry

Fluctuations in pupil diameter of the left eye were collected using an Eyelink 1000 (SR Research Ltd., Mississauga, Ontario, Canada), with a 1 kHz sampling frequency. Blinks, artefacts, and outliers were removed and linearly interpolated (*Wainstein et al., 2017*). High-frequency noise was smoothed using a second-order 2.5 Hz low-pass Butterworth filter. To obtain the pupil diameter average profile, data from each participant were normalised across each trial (corresponding to the 15 consecutive image set). This allowed us to correct for low-frequency baseline changes without eliminating the load effect and baseline differences due to load manipulations (*Campos-Arteaga et al., 2020*; *Rojas-Líbano et al., 2019*).

## Recurrent neural network modelling

We used PyTorch (*Paszke, 2019*) to implement and train 50 continuous-time RNNs that we constrained to respect Dale's law ($N_{E+I}$=40, 80% excitatory $N_E$= 32, and 20% inhibitory $N_I$= 8) using the procedure set out in *Yang and Wang, 2020*. The dynamics of each network evolved according to the following system of stochastic differential equations:

$$dx = \frac{1}{\tau} \left( -x\left(t\right) + W^{rec} r\left(t\right) + W^{in} u\left(t\right) \right) dt + dW$$

where $x \in \mathbb{R}^{N \times 1}$ represents the sub-threshold activation of each unit, $u \in \mathbb{R}^{2 \times 1}$ the external input into the network, $W^{rec} \in \mathbb{R}^{40 \times 40}$ the recurrent weights, $W^{in} \in \mathbb{R}^{40 \times 2}$ the input weights, and $\tau$ the time constant which we set to 100 ms. In addition to task input, each unit in the network was driven by a Weiner process $dW$. The subthreshold activation variable $x$ was converted into a vector of instantaneous firing rates by applying a sigmoid function $r = \frac{1}{1+e^{(-gx)}}$, where $g \in \mathbb{R}^{N \times 1}$ is a vector containing the gain control parameter of each unit's activation function that was multiplied element wise with $x$. Network outputs $z \in \mathbb{R}^{2 \times 1}$ were given by a linear readout of the excitatory population's firing rate

$z = W^{out} r_E$. Where $W^{out} \in \mathbb{R}^{N_E \times 2}$. The network's choice at each time point was the maximum of the two-dimensional output $z$.

We imposed Dale's law on the recurrent weights of the network by parametrising the weight matrix with a mask $W^{mask} \in \mathbb{R}^{40 \times 40}$, which contained zeros in the leading diagonal (removing self-connections), +1 in all non-diagonal entries of the first 32 rows/columns, and -1 in the remaining eight rows/columns. We obtained the constrained recurrent weight matrix by multiplying the absolute value of the trained weights element wise with the mask $W^{rec} = \left| W^{rec}_{plastic} \right| \odot W^{mask}$, thereby imposing an 80/20 E/I ratio. Similarly, we constrained the projection of the input to the network and the readout projection to be strictly positive by taking the absolute value of the trained input and output weights $W^{in} = \left| W^{in}_{plastic} \right|$, $W^{out} = \left| W^{out}_{plastic} \right|$.

Following standard practice (*Yang and Wang, 2020*), we simulated the network by discretising the system using a Euler–Maruyama integration scheme, where $\alpha = \frac{dt}{\tau}$, and $\sigma_{rec} = 0.01$.

$$x\left(t + \Delta t\right) = \left(1 - \alpha\right) x\left(t\right) + \alpha \left(W^{rec} r\left(t\right) + W^{in} u\left(t\right)\right) + \sigma_{rec} \sqrt{\Delta t} N\left(0, 1\right)$$

Each network was trained by optimising $W^{in}_{plastic}$, $W^{rec}_{plastic}$, and $W^{out}_{plastic}$ to minimise a cross-entropy loss function through 1000 iterations of back propagation through time (*Werbos, 1990*) with ADAM (*Kingma and Ba, 2015*). Batches consisted of single trials which for our simple task (described below) was sufficient for each network to converge on near-perfect behavioural accuracy. All training was performed with the gain control parameter set to 1. Again following standard practice, we trained the networks with a relatively large time step of $\Delta t$ = 200 ms. Following training, to ensure numerical stability we exported the trained weights into MATLAB and simulated the system with a bespoke numerical integration scheme with $\Delta t$ set 1ms.

The task consisted of a simple change detection paradigm analogous to the task performed by our human participants. Specifically, at each time point the network was fed a two-dimensional input $u(t) = \begin{bmatrix} u_1 & u_2 \end{bmatrix}^T$, with each column representing the 'sensory evidence' for each of the two stimulus categories. The task lasted for 1 s of simulation time beginning with maximum evidence for one of the two categories $u(t) = \begin{bmatrix} 1 & 0 \end{bmatrix}^T$ and over the course of each trial changed linearly such so that at the half-way point of the simulation the sensory evidence for each stimulus category changed was perfectly matched category $u(t) = \begin{bmatrix} .5 & .5 \end{bmatrix}^T$ and by the final time step consisted of maximum evidence for the second stimulus category $u(t) = \begin{bmatrix} 0 & 1 \end{bmatrix}^T$. We trained the network to output a response for stimulus category 1 whenever $u_1 > 0.5$, and $u_2 < 0.5$, and category 2 whenever $u_1 < 0.5$, and $u_2 > 0.5$.

To test our hypothesis that perceptual uncertainty increases neuromodulatory via phasic bursts in the noradrenergic LC, we made gain time dependent with dynamics governed by a linear ODE with a forcing term proportional to the uncertainty (i.e. the entropy $H\left(z\right) = \sum_i p\left(z\right)_i \ln\left(p\left(z\right)_i\right)$) of the network's readout.

$$dg = \frac{1}{\tau} \left(g_{tonic} - g\left(t\right) + \gamma H\left(z\right)\right) dt$$

where $p\left(z\right)$ is obtained by passing $z\left(t\right)$ through a softmax function at each time step of the simulation $p\left(z\right)_i = \frac{exp\left(\omega z_i\right)}{\sum_j^K exp\left(\omega z_j\right)}$ with inverse temperature parameter $\omega = 0.25$. When the network approaches the half-way point in the trial, input is maximally ambiguous and the distribution $p\left(z\right)$ approaches a uniform distribution leading $H\left(z\right)$ to approach its maximum value, which in turn leads to a phasic increase in gain (with magnitude $\gamma$). In the absence of forcing (i.e. under conditions of perceptual certainty), gain decays exponentially to its tonic value ($g_{tonic} = 1$).

To study how the population dynamics of the trained networks changed as a function of gain in a shared space, we performed a PCA on the concatenated activity of the network at $\gamma = 0$. The set of principal components was highly low-dimensional, with 80.58±6.34% of the variance explained by the first principal component (PC$_1$). We then projected the trial-averaged activity at each gain value at each timepoint onto the top PC.

## Energy landscape analysis

Leveraging previous work from our group (*Munn et al., 2021*), we constructed a measure of the energy landscape traversed by each network through an analogy to the relationship between probability and energy in statistical mechanics (*Tkačik et al., 2015*; *Munn and Gong, 2018*) given by the Boltzmann distribution.

$$p_i = \frac{1}{z}e^{-\beta E_i}$$

where $p_i$ denotes the probability of each state, $E_i$ the energy of each state, $\beta$ the thermodynamic beta, and $z$ the canonical partition function. Solving for $E_i$, we obtain

$$E_i = \frac{1}{\beta}ln\left(\frac{1}{zp_i}\right)$$

Instead of inferring the probability distribution from the energy of a state as is done in physics, we used the fitdist function in MATLAB with a Gaussian kernel ($P(x) = \frac{1}{4n}\sum_{i=1}^{n}K\left(\frac{x}{4}\right)$, where $K(u) = \frac{1}{2\sqrt{\pi}}e^{\frac{-1}{2}u^2}$) to infer the probability of the state, and then solved for the energy. As $\int_{-\infty}^{+\infty}P(x)\,dx = 1$ by construction, the partition function $z$, which we define here to be the integral of the pdf, is equal to 1, which, after setting $\beta = 1$, yields

$$E_i = ln\left(\frac{1}{p_i}\right)$$

For the allocentric landscape analysis, we defined the state of the system in terms of the trial-averaged loadings on $PC_1$ which we divided into 250 ms windows. For the egocentric landscape analysis, we calculated the mean-squared displacement (MSD) of the activity of the RNN at each time point $\tau_0$ relative to the reference point $\tau_0 + \tau$:

$$MSD_{\tau,\tau_0} = \left\langle\left|x_{\tau_0+\tau} - x_{\tau_0}\right|^2\right\rangle_n$$

For congruency with the allocentric analysis, we increased $\tau$ and $\tau_0$ in steps of 250 ms starting 1 s into the trial and ending with a maximum difference between $\tau$ and $\tau_0$ of 5 s to ensure that all steps had equivalent window sizes.

Following the physical analogy, we think of the state of the system, $PC_1$ loadings in the allocentric analysis, and MSD in the egocentric analysis, as akin to the location and movement of a particle respectively. Positions in state space with low energy have a higher probability of being occupied, and systems with a higher average energy have a more uniform probability distribution, making large jumps in the position of a particle more likely (i.e. lower energy for large MSD values; see supplementary material; *Figure 4—figure supplement 1*).

To quantify the effect of gain-mediated alterations to the topography of the allocentric energy landscape, we devised a novel measure – neural work – of the force (which in classical mechanics is equal to the negative gradient of potential energy) exerted on the low-dimensional neural trajectory by the vector field quantified by the allocentric energy landscape at each time point in the trial.

$$W_t = -\frac{dE_t}{dx}s_t$$

where $s_t$ is the displacement of the PC trajectory, and $\frac{dE_t}{dx}$ the energy gradient. We computed $s_t$ from the (absolute) difference between $PC_1$ loadings at the start and end of each time window, and $\frac{dE_t}{dx}$ from the gradient of energy values at the start and end of each time window.

## MRI data

Functional data were acquired using gradient echo-planar T2*-weighted images collected on a 1.5T Phillips scanner located at Grand River Hospital in Waterloo, Ontario (TR = 2000 ms; TE = 40 ms; slice thickness = 5 mm with no gap; 26 slices/volume; FOV = 220 × 220 mm²; voxel size = 2.75 × 2.75 × 5 mm³; flip angle = 90°). Each experimental run consisted of 26 slices per volume and 285 volumes. At the beginning of each run, a whole-brain T1-weighted anatomical image was collected for each

participant (TR = 7.4 ms; TE = 3.4 ms; voxel size = 1 × 1 × 1 mm³; FOV = 240 × 240 mm²; 150 slices with no gap; flip angle = 8°). The experimental protocol was programmed using E-Prime experimental presentation software (v1.1 SP3; Psychology Software Tools, Pittsburgh, PA). Stimuli were presented on an Avotec Silent Vision fibre-optic presentation system using binocular projection glasses (Model SV-7021). The onset of each trial was synchronised with the onset of data collection for the appropriate functional volume using trigger pulses from the scanner.

## fMRI data preprocessing

After realignment (using FSL's MCFLIRT), we used FEAT to unwarp the EPI images in the y-direction with a 10% signal loss threshold and an effective echo spacing of 0.333. Following noise-cleaning with FIX (custom training set for scanner, threshold 20, including regression of estimated motion parameters), the unwrapped EPI images were then smoothed at 6 mm FWHM and nonlinearly co-registered with the anatomical T1 to 2 mm isotropic MNI space. Temporal artefacts were identified in each dataset by calculating framewise displacement (FD) from the derivatives of the six rigid-body realignment parameters estimated during standard volume realignment (*Power et al., 2014*), as well as the root mean square change in BOLD signal from volume to volume (DVARS). Frames associated with FD > 0.25 mm or DVARS > 2.5% were identified; however, as no participants were identified with greater than 10% of the resting time points exceeding these values, no trials were excluded from further analysis. There were no differences in head motion parameters between the five runs (p>0.500). Following artefact detection, nuisance covariates associated with the six linear head movement parameters (and their temporal derivatives), DVARS, physiological regressors (created using the RETROICOR method), and anatomical masks from the cerebrospinal fluid and deep cerebral white matter were regressed from the data using the CompCor strategy (*Behzadi et al., 2007*). Finally, in keeping with previous time-resolved connectivity experiments (*Gu et al., 2015*), a temporal band pass filter (0.0071<f<0.125 Hz) was applied to the data.

## Brain parcellation

Following preprocessing, the mean time series was extracted from 375 predefined regions of interest (ROIs). To ensure whole-brain coverage, we extracted the following: (a) 333 cortical parcels (161 and 162 regions from the left and right hemispheres, respectively) using the Gordon atlas (*Gordon et al., 2016*); (b) 14 subcortical regions from the Harvard-Oxford subcortical atlas (bilateral thalamus, caudate, putamen, ventral striatum, globus pallidus, amygdala, and hippocampus; https://fsl.fmrib.ox.ac.uk/); and (c) 28 cerebellar regions from the SUIT atlas (*Diedrichsen et al., 2009*) for each participant in the study.

## Neuroimaging analysis

In order to analyse task-evoked activity related to stimulus presentations, we first performed a PCA (*Shine et al., 2019*) on the pre-processed BOLD time series (per subject/session) to extract orthogonal low-dimensional time series. The top 3 PCs explained ~30.6% of the variance. The time series of these PCs was entered into a general linear model, in which we modelled the following nine event types across an entire session, centred around the perceptual switch point, which changed on a trial-by-trial basis: the first two images (modelled as a single regressor), the seven images surrounding each perceptual change (i.e. the switch trial and the three images surrounding the change point, modelled as seven separate regressors) and the last two images (modelled as a single regressor). Each of the event onset times was also convolved with a canonical haemodynamic response function. This left us with nine unique β values per principal component, which we could use to determine how each PC differentially engaged as a function of the task. To test the hypothesis that the rate of change of PC engagement peaked at the perceptual change point, we calculated the difference between the β value for each of the top 3 PCs for each of the nine event types, and then plotted the resultant series in order to identify whether a peak occurred at the perceptual switch point (i.e. the middle β value in the series). A block-resampling null (n=5000 permutations) was used as a permutation test (p<0.05).

Spatial maps associated with the terms 'switching', 'effort', 'attention', 'perception', and 'load' were downloaded from the *neurosynth* repository (*Yarkoni et al., 2011*) and mapped into our parcellation space by calculating the mean value within each independent parcel. These values were then correlated with the spatial loading of each of the top 3 PCs. A separate partial correlation analysis was

conducted in which the same correlation was estimated after controlling for each of the other spatial maps.

## Topological analyses

A hierarchical modularity approach was used to collapse the mean time-averaged correlation matrix across participants into a set of four spatially non-overlapping modules. Briefly, this involved running the Louvain modularity algorithm, which iteratively maximises the modularity statistic, $Q$, for different community assignments until the maximum possible score of $Q$ has been obtained.

$$Q_T = \frac{1}{v^+} \sum_{ij} (w_{ij}^+ - e_{ij}^+)\delta_{M_i M_j} - \frac{1}{v^+ + v^-} \sum_{ij} (w_{ij}^- - e_{ij}^-)\delta_{M_i M_j}$$

The community assignment for each region was then estimated 500 times across a range of γ values (0.5–2.0, in steps of 0.1). In order to identify multi-level structure in our data, we repeated the modularity analysis for each of the modules identified in the first step (*Meunier et al., 2010*). Finally, a consensus partition was identified using a fine-tuning algorithm from the Brain Connectivity Toolbox (http://www.brain-connectivity-toolbox.net/). We subsequently used this final module assignment to estimate the cartographic profile of the each participant's time-averaged adjacency matrix (*Shine et al., 2016*). Specifically, we estimated integration using the participation coefficient, which quantifies the extent to which a region connects across all modules (i.e. between-module strength; *Guimerà and Nunes Amaral, 2005*), and segregation using the module-degree Z-score. These measures were entered into a joint-histogram (101 × 101 unique bins, equally spaced between 0 and 1 [for integration] and –1 and 1 [for segregation]). The value within each bin of this joint histogram was then correlated with the combined regression weights of $PC_2$ and $PC_3$ for each subject. A permutation test that scrambled the order of participants was used to assess statistical significance ($p<0.05$).

## Brain-state displacement and the energy landscape

To quantify the change in the evoked BOLD activity following each stimulus, we calculated the main BOLD displacement (MBD). The MBD is a measure of the absolute evoked deviation in BOLD activity. The evoked activity is measured through a general linear model using a canonical haemodynamic response function convolved on a design matrix. We are interested in the probability, $p$(MBD, $re$), that we will observe a given displacement in BOLD at a given regressor $re$. The probability is calculated through the null model of the general linear model (the probability that the observed evoked value of the corresponding region is different from 0). As described above, we then calculated the energy for each displacement value as $E_{MBD,re} = ln\left(\frac{1}{P(MBD,re)}\right)$. Finally, to measure the surprise per displacement, we divided the absolute β for PC2 from $E_{MBD,re}$ for each regressor $re$ (*Figure 6*).

## Additional information

### Funding

| Funder | Grant reference number | Author |
|---|---|---|
| National Health and Medical Research Council | | James M Shine |
| Australian Research Council | | James M Shine |

The funders had no role in study design, data collection and interpretation, or the decision to submit the work for publication.

### Author contributions

Gabriel Wainstein, Conceptualization, Data curation, Formal analysis, Validation, Investigation, Visualization, Methodology, Writing – original draft, Writing – review and editing; Christopher J Whyte, Conceptualization, Data curation, Software, Formal analysis, Validation, Investigation, Visualization, Methodology, Writing – original draft, Writing – review and editing; Kaylena A Ehgoetz Martens,

Conceptualization, Writing – review and editing; Eli J Müller, James Danckert, Methodology, Writing – review and editing; Vicente Medel, Writing – review and editing; Britt Anderson, Data curation, Methodology, Writing – review and editing; Elisabeth Stöttinger, Data curation, Writing – review and editing; Brandon R Munn, Conceptualization, Formal analysis, Supervision, Investigation, Methodology, Writing – review and editing; James M Shine, Conceptualization, Resources, Formal analysis, Supervision, Validation, Investigation, Visualization, Methodology, Writing – original draft, Project administration, Writing – review and editing

### Author ORCIDs
Gabriel Wainstein ⓘ https://orcid.org/0000-0002-8106-6647
Christopher J Whyte ⓘ https://orcid.org/0000-0002-4627-0503
Kaylena A Ehgoetz Martens ⓘ https://orcid.org/0000-0002-8488-2295
Eli J Müller ⓘ https://orcid.org/0000-0003-2497-0194
Vicente Medel ⓘ https://orcid.org/0000-0003-2443-8683
Elisabeth Stöttinger ⓘ https://orcid.org/0000-0003-4544-8944
James Danckert ⓘ https://orcid.org/0000-0001-8093-066X
Brandon R Munn ⓘ https://orcid.org/0000-0002-3638-1605
James M Shine ⓘ https://orcid.org/0000-0003-1762-5499

### Ethics
The research protocol was approved by the Office of Research Ethics at the University of Waterloo and the Tri-Hospital Research Ethics Board of the Region of Waterloo in Ontario, Canada.

Reviewer #1 (Public review): https://doi.org/10.7554/eLife.93191.4.sa1
Reviewer #2 (Public review): https://doi.org/10.7554/eLife.93191.4.sa2
Author response https://doi.org/10.7554/eLife.93191.4.sa3

---

## Additional files

### Supplementary files
MDAR checklist

### Data availability
The data and code necessary to replicate our results are available online (https://github.com/ShineLabUSYD/AmbiguousFigures [copy archived at *Whyte et al., 2025*] and https://osf.io/uvykp/).

The following dataset was generated:

| Author(s) | Year | Dataset title | Dataset URL | Database and Identifier |
|---|---|---|---|---|
| Danckert J | 2025 | fMRI data from Stottinger papers | https://doi.org/10.17605/OSF.IO/UVYKP | Open Science Framework, 10.17605/OSF.IO/UVYKP |

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
