## [Editor Report · eLife Assessment]

This **valuable** article explores the idea that transient modulations of neural gain promote switches between distinct perceptual interpretations of ambiguous stimuli. The authors provide **solid** evidence for this idea by pupillometry (an indirect proxy of neuromodulatory activity), fMRI, neural network modelling, and dynamical systems analyses. The highly integrative nature of this approach is rare in the field.

---

## [Referee Report · Reviewer #1 (Public review)]

Summary:

This paper proposes a neural mechanism underlying the perception of ambiguous images: neuromodulation changes the gain of neural circuits promoting a switch between two possible percepts. Converging evidence for this is provided by indirect measurements of neuromodulatory activity and large-scale brain dynamics which are linked by a neural network model. However, both the data analysis as well as the computational modeling are incomplete and would benefit from a more rigorous approach.

This is a revised version of the manuscript which, in my view, is a considerable step forward compared to the original submission.

In particular, the authors now model phasic gain changes in the RNN, based on the network's uncertainty. This is original and much closer to what is suggested by the phasic pupil responses. They also show that switching is actually a network effect because switching times depend on network configuration (Fig 2). This resolves my main comments 1 and 2 about the model.

The mechanism, as I understand it, is different from what the authors described before in the RNN with tonic gain changes. As uncertainty increases, the network enters a regime in which the two excitatory populations start to oscillate. My intuition is that this oscillation arises from the feedback loop created by the new gain control mechanism. If my intuition is correct, I think it would be worth to explain this mechanism in the paper more explicitly.

Comments on revisions:

This is a second revision. I have no further comments. The authors have not answered the question that I had in the previous round (about the origin of oscillations in the RNN). I think this topic deserves to be explored in more detail but perhaps that is beyond the scope of the current paper.

---

## [Referee Report · Reviewer #2 (Public review)]

This paper tests the hypothesis that perceptual switches during the presentation of ambiguous stimuli are accompanied by changes in neuromodulation that alter neural gain and trigger abrupt changes in brain activity. To test this hypothesis, the study combines pupillometry, artificial recurrent network (RNN) analysis and fMRI recording. In particular, the study uses methods of energy landscape analysis inspired by physics, which is particularly interesting.

Strengths

- The authors should be commended for combining different methods (pupillometry, RNNs, fMRI) to test their hypothesis. This combination provides a mechanistic insight into perceptual switches in the brain and artificial neural networks.

- The study combines different viewpoints and fields of scientific literature, including neuroscience, psychology, physics, dynamical systems. In order to make this combination more accessible to the reader, the different aspects are presented in a pedagogical way to be accessible to all fields.

- This combination of methods and viewpoints is rarely done, so it is very useful.

- The authors introduce dynamic gain modulation in their recurrent neural network, which is novel. They devote a section of the paper to studying the dynamics, fixed points and convergence of this type of network.

Weaknesses

- The study may not be specific to perceptual switches. This is because the study relies on a paradigm in which participants report when they identify a switch in the item category. Therefore, it is unclear whether the effects reported in the paper are related to the perceptual switch itself, to attention, or to the detection of behaviourally relevant events. The authors are cautious and explicitly acknowledge this point in their study.

- The demonstration of the causal role of gain modulation in perceptual switches is partial. This causality is clearly demonstrated in the simulation work with the RNN. However, it is not fully demonstrated in the pupil analysis and the fMRI analysis. One reason is that this work is correlative (which is already very informative).

- Some effects may reflect the expectation of a perceptual switch rather than the perceptual switch itself. To mitigate this risk, the design of the fMRI task included catch trials, in which no switch occurs, to reduce the expectation of a switch. The pupil study, however, did not include such catch trials.

- The paper uses RNN-based modelling to provide mechanistic insight into the role of gain modulation in perceptual switches. However, the RNN solves a task that differs from that performed by human participants, which may limit the explanatory value of the model. The RNN is provided with two inputs characterising the sensory evidence supporting the first and last image category in the sequence (e.g. plane and shark). In contrast, observers in the task don't know in advance the identity of the last image at the beginning of the sequence. The brain first receives sensory evidence about the image category (e.g. plane) with which the sequence begins, which is very easy to recognise, then it sees a sequence of morphed images and has to discover what the final image category will be. To discover the final image category, the brain considers several possibilities for the second images (it is a shark?, a frog?, a bird?, etc.), rather than comparing the likelihood of just two categories. This search process among many alternatives and the perceptual switch in the task is therefore different from the competition between only two inputs in the RNN.

- Another aspect of the motivation for the RNN model remains unclear. The authors introduce dynamic gain modulation in the RNN, but it is not clear what the added value of dynamic gain modulation is. Both static (Fig. S1) and dynamic (Fig. 2F) gain modulation lead to the predicted effect: faster switching when the gain is larger.

- The authors are to be commended for addressing their research questions with multiple tools and approaches. There are links between the different parts of the study. The RNN and the pupil are linked by the notion of gain modulation, the RNN and the fMRI analysis are linked by the study of the energy landscape, the fMRI study and the pupil study are indirectly linked by previous work for this group showing that the peak in LC fMRI activity precedes a flattening of the energy landscape. These links are very interesting but could have been stronger and more complete.

Comments on revisions:

I thank the authors for their responses.

My review presents points that the authors themselves present as weaknesses or limitations. It also includes points that cannot be addressed in a revision (e.g. causality).

Regarding the fact that the RNN only considers two categories, whereas subjects consider more categories (because they don't know the final image), I have toned down my remark (removing "markedly" different, removing the fact that the hypothesis space is vast given that participants have some priors). I also removed the qualifier "mechanistically" different, because it can be understood in different ways. The point remains that the proposed model has 2 inputs, the corresponding network in the brain has >2 inputs (because it considers more categories than the RNN), which is different, and which is the point of my remark. I think it may limit the value of the model, but I don't think it is not "sensible".

---

## [Author Response]

The following is the authors’ response to the previous reviews

**Reviewer #1 (Public r**eview):The mechanism, as I understand it, is different from what the authors described before in the RNN with tonic gain changes. As uncertainty increases, the network enters a regime in which the two excitatory populations start to oscillate. My intuition is that this oscillation arises from the feedback loop created by the new gain control mechanism. If my intuition is correct, I think it would be worth to explain this mechanism in the paper more explicitly.

While interesting, this intuition is not correct. The oscillations are generated by the interaction between excitatory and inhibitory nodes in the network and occur in the model even with stationary gain. All of the plots in figure 3 exploring the dynamical regime of the network at different input x gain combinations (i.e., where the oscillatory regime is characterised) are simulations run with stationary gain.

To ensure that this intuition is more clearly presented in the manuscript, we have edited the description in the text.

P. 12: “Because of the large size of the network, we could not solve for the fixed points or study their stability analytically. Instead, we opted for a numerical approach and characterised the dynamical regime (i.e. the location and existence of approximate fixed-point attractors) across all combinations of (static) gain and visited by the network.”

**Reviewer #2 (Public review):**
- The demonstration of the causal role of gain modulation in perceptual switches is partial. This causality is clearly demonstrated in the simulation work with the RNN. However, it is not fully demonstrated in the pupil analysis and the fMRI analysis. One reason is that this work is correlative (which is already very informative). An analysis of the timing of the effect might have overcome this limitation. For example, in a previous study, the same group showed that fMRI activity in the LC region precedes changes in the energy landscape of fMRI dynamics, which is a step towards investigating causal links between gain modulation, changes in the energy landscape and perceptual switches.

Thank you for the suggestion, which we considered in detail. Unfortunately, the temporal and spatial resolution of the fMRI data collected for this study precluded the same analyses we’ve run in previous work, however this is an important question for future work.

- Some effects may reflect the expectation of a perceptual switch rather than the perceptual switch itself. To mitigate this risk, the design of the fMRI task included catch trials, in which no switch occurs, to reduce the expectation of a switch. The pupil study, however, did not include such catch trials.

We agree that this is a limitation of the current study, which we previously highlighted in the methods section.

- The paper uses RNN-based modelling to provide mechanistic insight into the role of gain modulation in perceptual switches. However, the RNN solves a task that differs markedly from that performed by human participants, which may limit the explanatory value of the model. The RNN is provided with two inputs characterising the sensory evidence supporting the first and last image category in the sequence (e.g. plane and shark). In contrast, observers in the task were naïve as to the identity of the last image at the beginning of the sequence. The brain first receives sensory evidence about the image category (e.g. plane) with which the sequence begins, which is very easy to recognise, then it sees a sequence of morphed images and has to discover what the final image category will be. To discover the final image category, the brain has to search a vast space of possible second images (it is a shark?, a frog?, a bird?, etc.), rather than comparing the likelihood of just two categories. This search process and the perceptual switch in the task appear to be mechanistically different from the competition between two inputs in the RNN.

We appreciate the critical analysis of the experimental paradigm but disagree with the reviewers conclusions for two keys reasons: (1) Participants prior exposure to the images, such that they could create an expectation about what stimulus category a particular image would transition into (i.e., the image could not switch into any possible category); and (2) even if the reviewers’ concern was founded, models of *K* winner-take-all decision making are structured identically irrespective of whether the options are 2 or *K* options all that changes is the simulated reaction times which depend linearly on the K (for an example model see Hugh Wilson’s textbook Spikes, Decisions, and Actions, 1999, p.89-91). For these reasons, we maintain that the RNN is a sensible representation of the behavioural task.

- Another aspect of the motivation for the RNN model remains unclear. The authors introduce dynamic gain modulation in the RNN, but it is not clear what the added value of dynamic gain modulation is. Both static (Fig. S1) and dynamic (Fig. 2F) gain modulation lead to the predicted effect: faster switching when the gain is larger.

While we agree that the effect is observable with both static and dynamic gain, the stronger construct validity associated with the dynamic approach, including a stronger link with the observed pupil dynamics and a rich literature associated with modelling the behavioural consequences of surprise/uncertainty led us to the conclusion that the dynamical approach was a better representation of our hypothesis.

- Fig 1C: I don't see a "top grey bar" indicating significance.

Thank you for catching this, the caption has been amended. The text was from an older version of the manuscript.

- p. 10, reference to fig 3F seems incorrect: there is Fig 3F upper and Fig 3F lower, and nothing on Fig 3 and its legend mention the lesion of units

This has been amended. We meant to refer to 2F.

- In the response letter you mention a MATLAB tutorial, but I could not find it.

This has been amended. Github repository can be found at https://github.com/ShineLabUSYD/AmbiguousFigures